# Unconventional Dual Donor-Acceptor Topologies of Aromatic Rings in Amine-Based Polymeric Tetrahedral Zn(II) Compounds Involving Unusual Non-Covalent Contacts: Antiproliferative Evaluation and Theoretical Studies

Pranay Sharma [1], Rosa M. Gomila [2], Miquel Barceló-Oliver [2], Akalesh K. Verma [3],*, Diksha Dutta [3], Antonio Frontera [2],* and Manjit K. Bhattacharyya [1],*

[1] Department of Chemistry, Cotton University, Guwahati 781001, Assam, India
[2] Departament de Química, Universitat de les Illes Balears, Crta de Valldemossa km 7.7, 07122 Palma de Mallorca (Baleares), Spain
[3] Cell & Biochemical Technology Laboratory, Department of Zoology, Cotton University, Guwahati 781001, Assam, India
* Correspondence: akhilesh@cottonuniversity.ac.in (A.K.V.); toni.frontera@uib.es (A.F.); manjit.bhattacharyya@cottonuniversity.ac.in (M.K.B.)

**Abstract:** Two Zn(II) coordination polymers, viz., $[Zn_2Cl_2(H_2O)_2(\mu\text{-}4\text{-}AmBz)_2]_n$ (**1**) and $[ZnCl_2(\mu\text{-}3\text{-}AmPy)_2]_n$ (**2**) (4-AmBz = 4-aminobenzoate, 3-AmPy = 3-aminopyridine) have been prepared at room temperature and characterized using elemental analysis, FT-IR, electronic spectroscopy, TGA (thermogravimetric analysis) and single crystal XRD. Crystal structure analyses of the polymers unfold the presence of non-covalent anion–π, π-stacking and unusual $NH_2(amino)\cdots\pi$ interactions which provide rigidity to the crystal structures. Unconventional Type I Cl···Cl interactions also play a pivotal role in the stability of compound **1**. Molecular electrostatic potential (MEP) surface analysis reveals that the MEP values over the center of the aromatic rings of coordinated *4-AmBz* and *3-AmPy* moieties are positive on one side and negative on the other side which confirms the dual non-covalent donor-acceptor topologies of the aromatic rings and explains the concurrent formation of unusual non-covalent $NH_2\cdots\pi$ and anion–π interactions. DFT (density functional theory) calculations, QTAIM (quantum theory of atoms in molecules) and NCI plot (non-covalent index) index analyses reveal that among various non-covalent contacts involved in the crystal packing of the compounds, H-bonds in compound **1** and π-interactions ($NH_2\cdots\pi$, π-π, anion–π) in compound **2** are energetically significant. We have explored in vitro cytotoxic potential of the compounds in Dalton's lymphoma (DL) cancer cells using trypan blue and apoptosis assays. The studies show that compounds **1** and **2** can significantly exhibit cytotoxicity in DL cells with minimum cytotoxicity in healthy PBMC cells. Molecular docking studies reveal that the compounds effectively bind with the antiapoptotic target proteins; thereby establishing a structure activity relationship of the compounds.

**Keywords:** Zn(II) coordination polymers; dual donor-acceptor topologies; unusual non-covalent contacts; cytotoxicity; docking

## 1. Introduction

Coordination compounds with organic bridging ligands have gained appreciable research interest not only due to their intriguing structural topologies but also because of their myriad potential applications in biological systems, luminescence, catalysis, drug development, non-linear optics, magnetism, etc. [1–5]. Looking at the potential applications of poly-nuclear compounds, various coordination compounds with one-dimensional polymeric structures have been developed [6,7]. However, it is still a difficult task for the researchers to design the architectures with desired structural topologies as the self-assembly of molecular entities mainly depends on a variety of experimental conditions,

viz., metal-to-ligand ratio, pH of the medium, solvents, temperature, and coordination geometry of the metal centers, among others [8].

Studies on non-covalent interactions such as hydrogen bonding, electrostatic and charge-transfer interactions, π-stacking and metal ion coordination, etc. have attracted researchers from the crystal engineering viewpoint [9–11]. Hydrogen bonding—one of the commonly observed non-covalent interactions—can effectively control the secondary coordination sphere of metal ions [12,13]. Supramolecular non-covalent contacts involving the aromatic π-systems have also played important roles in the stability of the crystal structures [14]. It has been well established that metal-coordinated organic moieties with aromatic rings can be either electron-rich (donor) or electron-deficient (acceptor) to direct molecular self-assemblies with diverse structural topologies [15,16]. However, dual non-covalent donor-acceptor topologies of aromatic rings of metal-coordinated organic ligands are still only scarcely explored in the literature. $Cl \cdots Cl$ interaction has also emerged as a potential non-covalent interaction which can also be termed a donor-acceptor interaction [17]. Moreover, unusual $NH_2 \cdots \pi$ type non-covalent interactions have also gained interest from researchers in the field of supramolecular chemistry [18]. Experimental analysis aided by computational studies is important to visualize such unusual non-covalent interactions which can be used for engineering crystal structures with desired physical and chemical properties.

Aromatic organic donors with carboxyl groups can be used as potential candidates to develop coordination compounds with multi-dimensional assemblies because of the flexible coordination motifs [19,20]. Pyridine-based donors have also prompted immense research interest in coordination chemistry due to their potential applications in various fields [21,22]. Coordination compounds of aminopyridines have shown a wide range of biological applications [23]. Zinc is the second most abundant human body trace element which can protect against tissue damage [24,25]. Zinc plays a key role in cell proliferation as well as in DNA/RNA synthesis; hence, a deficiency of zinc can cause depression in the immune system [26]. Coordination compounds of zinc involving benzoate and substituted benzoate derivatives have been reported to possess interesting structural topologies [27,28]. Zinc compounds have also been reported to show significant pharmacological properties such as anticancer [29], antidiabetic [30], anti-inflammatory [31], antimicrobial [32–35], antioxidant [36], and in Alzheimer's disease [37]. There are several reports of anticancer activities of coordination compounds against DL cancer cells [38,39]. Jayendran and Kurup have reported a Zn(II) coordination compound involving NNO donor Schiff base ligand and explored its significant in vitro anticancer activity in the DL cancer cell line [40].

To explore the self-assembly, unconventional structural topologies, and anticancer properties of metal-organic polymers; in the present study, we have reported the synthesis and crystal structures of two Zn(II) coordination polymers, viz., $[Zn_2Cl_2(H_2O)_2(\mu\text{-}4\text{-}AmBz)_2]_n$ (**1**) and $[ZnCl_2(\mu\text{-}3\text{-}AmPy)_2]_n$ (**2**). The compounds have been further characterized using spectroscopic techniques (FT-IR and electronic), elemental analysis, and TGA. Crystal structure analysis of compound **1** reveals the presence of Type I $Cl \cdots Cl$ and unusual $NH_2(amino) \cdots \pi$ interactions which stabilize the crystal structure. Similarly, $NH_2(amino) \cdots \pi$, anion $\cdots \pi$, and π-π interactions stabilize the 2D assembly of compound **2**. We have explored the energetic features of the unusual supramolecular assemblies observed in compounds **1** and **2** using DFT studies. The interactions have been further characterized by using several computational tools such as molecular electrostatic potential (MEP) surface, non-covalent index (NCI) plot, and quantum theory of atoms in molecules (QTAIM). The cytotoxic potential of the compounds has been investigated using Trypan blue and apoptosis assays in the DL cancer cell line. Molecular docking studies of the compounds were performed to explore the possible interactions of the compounds with the target proteins which are related to cancer growth and progression.

## 2. Experimental Section

### 2.1. Materials and Methods

The chemicals required for the synthesis of the compounds, viz., anhydrous zinc(II) chloride, 4-aminobenzoic acid, and 3-aminopyridine were bought from Sigma Aldrich and used as received. The single crystal X-ray data of the compounds were recorded using D8 Venture diffractometer, having a Photon III 14 detector and Incoatec high brilliance IµS DIAMOND Cu tube equipped with an Incoatec Helios MX multilayer optics. The data collection of the compounds was carried out at 100 K. The elemental analysis of the compounds was performed using Perkin Elmer 2400 Series II CHNS/O analyzer. We have recorded the KBr phase FT-IR spectra of the compounds using Bruker alpha (II) infrared spectrophotometer in the frequency range 4000–500 $cm^{-1}$. Electronic spectra of the compounds were recorded using Shimadzu UV-2600 spectrophotometer. $BaSO_4$ powder was used as reference to record the UV-Vis-NIR spectra. $^1$H-NMR spectra of the compounds were recorded using Avance III HD 400 NMR spectrometer with DMSO-d6 as solvent and tetramethylsilane (TMS) as the chemical shift reference. Thermal studies of the compounds were carried out using Mettler Toledo TGA/DSC1 STAR$^e$ system with the flow of $N_2$ gas at the heating rate of 10 °C $min^{-1}$.

### 2.2. Syntheses

#### 2.2.1. Synthesis of $[Zn_2Cl_2(H_2O)_2(\mu\text{-}4\text{-}AmBz)_2]_n$ (1)

Anhydrous $ZnCl_2$ (0.136 g, 1 mmol) and sodium salt of 4-aminobenzoic acid (0.159 g, 1 mmol) were mixed in 10 mL of de-ionized water and mechanically stirred at room temperature for two hours (Scheme 1). After two hours, the colourless resulting solution was kept unperturbed in cooling conditions (2–4 °C) for crystallization. After a few days, block-shaped colorless crystals suitable for single-crystal XRD were obtained from the mother liquor. Yield: 0.444 g (87%). Anal. calcd. for $C_{14}H_{16}N_2O_6Cl_2Zn_2$: C, 32.97%; H, 3.16%; N, 5.49%; Found: C, 32.88%; H, 3.07%; N, 5.39%. FT-IR (KBr pellet, $cm^{-1}$): 3448(br), 2077(w), 1621(s), 1428(w), 1388(s), 1356(sh), 1220(w), 1181(w), 1141(w), 1067(m), 862(w), 783(m), 699(w), 679(w) (s, strong; m, medium; w, weak; br, broad; sh, shoulder).

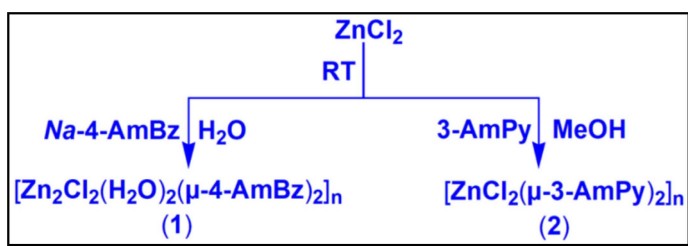

**Scheme 1.** Synthesis of compounds **1** and **2**.

#### 2.2.2. Synthesis of $[ZnCl_2(\mu\text{-}3\text{-}AmPy)_2]_n$ (2)

A mixture of anhydrous $ZnCl_2$ (0.136 g, 1 mmol) and 3-aminopyridine (0.188 g, 2 mmol) was dissolved in methanol (10 mL) and mechanically stirred at room temperature for two hours (Scheme 1). The resulting colorless solution was then kept unperturbed in a refrigerator (below 4 °C) for crystallization. After several days, colorless single crystals were obtained from the mother liquor. Yield: 0.196 g (85%). Anal. calcd. for $C_5H_6N_2Cl_2Zn$: C, 26.06%; H, 2.62%; N, 12.16%; Found: C, 25.97%; H, 2.57%; N, 12.08%. FT-IR (KBr pellet, $cm^{-1}$): 3448(br), 3258(w), 3129(w), 2108(w), 1639(s), 1555(sh), 1484(w), 1454(m), 1408(m), 1339(w), 1255(w), 1195(w), 1097(w), 1057(m), 1029(sh), 905(w), 816(w), 692(m), 651(m)

### 2.3. Crystallographic Data Collection and Refinement

The selected single crystals of **1** and **2** were covered with Parabar 10,320 (formally known as Paratone N) and mounted on a cryoloop on a D8 Venture diffractometer, with a Photon III 14 detector, using an Incoatec high brilliance IµS DIAMOND Cu tube equipped with an Incoatec Helios MX multilayer optics. The data collections of the compounds were

carried out at 100 K. Data reduction and cell refinements were performed using the Bruker APEX3 program [41]. SADABS program was used for scaling and absorption corrections in all cases [41]. The crystal structures of the compounds were solved by direct method and refined by full-matrix least-squares techniques with SHELXL-2018/3 [42] using WinGX [43] software. All non-hydrogen atoms were refined with anisotropic thermal parameters by full-matrix least-squares calculations on $F^2$. Hydrogen atoms were inserted at calculated positions and refined as riders. The structures were checked for higher symmetry with the help of the program PLATON [44]. The graphical materials have been prepared with the help of Mercury software [45]. Diamond 3.2 software is used to draw the molecular structures and the packing diagrams [46]. Collected data and refinement parameters for compounds **1** and **2** are summarized in Table 1.

**Table 1.** Crystallographic data and structure refinement details for compounds **1** and **2**.

| Parameters | 1 | 2 |
|---|---|---|
| Formula | $C_{14}H_{16}N_2Zn_2O_6Cl_2$ | $C_5H_6Cl_2ZnN_2$ |
| Formula weight | 509.93 | 230.39 |
| Temp, [K] | 294 | 100.0 |
| Crystal system | Monoclinic | Monoclinic |
| Space group | $P2/c$ | $Cc$ |
| a, [Å] | 15.4989(15) | 13.6324(10) |
| b, [Å] | 4.7429(5) | 7.5986(6) |
| c, [Å] | 12.2661(12) | 8.3812(6) |
| $\alpha$, [°] | 90 | 90 |
| $\beta$, [°] | 109.053(3) | 116.926(2) |
| $\gamma$, [°] | 90 | 90 |
| V, [Å$^3$] | 851.78(15) | 774.07(10) |
| Z | 2 | 4 |
| Absorption coefficient (mm$^{-1}$) | 6.692 | 10.113 |
| F(0 0 0) | 512.0 | 456.0 |
| $\varrho_{calc}$g/cm$^3$ | 1.988 | 1.977 |
| index ranges | $-18 \leq h \leq 18$ $-5 \leq k \leq 5,$ $-14 \leq l \leq 14$ | $-16 \leq h \leq 15,$ $-9 \leq k \leq 9,$ $-9 \leq l \leq 9$ |
| Crystal size, [mm$^3$] | $0.14 \times 0.12 \times 0.08$ | $0.32 \times 0.23 \times 0.17$ |
| 2θ range, [°] | 6.6036 to 137.118 | 13.744 to 133.266 |
| Independent Reflections | 1553 | 1330 |
| Reflections collected | 18,241 | 4757 |
| Refinement method | Full-matrix least-squares on $F^2$ | Full-matrix least-squares on $F^2$ |
| Data/restraints/parameters | 1553/0/120 | 1330/2/93 |
| Goodness-of-fit on $F^2$ | 1.236 | 1.072 |
| Final R indices [I > 2σ(I)] indices (all data) | $R_1 = 0.0554$, $wR_2 = 0.1645$ $R_1 = 0.0554$, $wR_2 = 0.1645$ | $R_1 = 0.0282$, $wR_2 = 0.0729$ $R_1 = 0.0282$, $wR_2 = 0.0729$ |
| Largest hole and peak [e·Å$^{-3}$] | $1.22/-0.76$ | $0.62/-0.67$ |

CCDC 2123028 and 2123034 contain the supplementary crystallographic data for compounds **1** and **2**, respectively. These data can be obtained free of charge at http://www.ccdc.cam.ac.uk (accessed on 22 February 2023) or from the Cambridge Crystallographic Data Centre, 12 Union Road, Cambridge CB2 1EZ, UK; fax: (+44) 1223-336-033; or E-mail: deposit@ccdc.cam.ac.uk.

*2.4. Computational Methods*

The theoretical study reported herein was performed using RI-BP86-D3/def2-TZVP [47,48] level of theory. For the calculations, the X-ray coordinates were used by means of the program Turbomole 7.2 [49] since we are interested in evaluating the noncovalent contacts as they stand in the solid state. The NCI plot [50] via reduced density gradient (RGD) isosurfaces and

QTAIM [51] methods were used to characterize the non-covalent interactions at the same level of theory. The MULTIWFN program [52] was for the calculation of the NCIplot isosurfaces and critical points. They were represented using the VMD software [53].

### 2.5. Cell Line and Drug Preparation

To explore in vitro anticancer activities of the compounds, Dalton's lymphoma (DL) cancer cells were used. DL is a malignant, transplantable T-cell lymphoma cell line; which has been effectively used in cancer research [54]. When choosing a cell line as the model system; the cell line's genetic stability and heterogeneity, host animal immunogenicity, and biological endpoints are always considered. In this regard, Dalton's lymphoma has evolved as an excellent model system where active components of a large number of natural plant products have been studied with biological endpoints [55]. It has been well established that peripheral blood mononuclear cells (PBMCs) are potential cell models that can be used to screen and investigate the effects of different synthesized compounds or drug molecules in the context of cancer research [56]. To explore the cytotoxicity of the compounds in normal cells; peripheral blood mononuclear cells (PBMC) were used. RPMI media was used to grow the DL cells which are used with a blood product, 10% FBS, streptomycin (100 g/mL), and penicillin (100 U/mL). The media was kept at 37 °C in $CO_2$ incubator with 5% $CO_2$. 50 mg of the compounds was dissolved in 1 mL of phosphate-buffered saline (PBS)/dimethyl sulfoxide (DMSO) (pH = 7.4) to prepare the stock solution of the compounds and then diluted to prepare the required solutions (0.01, 0.1, 0.5, 1, 5 and 10 μM).

### 2.6. Cytotoxicity and Apoptosis Assays

Trypan blue exclusion assay was performed in DL cancer cell line to evaluate the cytotoxic potential of the synthesized compounds [57]. Trypan blue dye is unable to penetrate the cell membrane of living cells but can go inside dead cells having damaged cell membranes [58]. Various concentrations (0.01, 0.1, 0.5, 1, 5, and 10 μM) of the compounds were used to explore the short-term (24 h) cytotoxic potential in 96 cell culture plates (Thermo Scientific, Cat. No: 265301). $IC_{50}$ (concentration required for 50% cell death) values of the compounds in DL and PBMC cell lines were determined using the dose response-linear curve fit method. For DL and PBMC cell lines; 0.01–100 and 0.01–400 μM concentration ranges were used, respectively. The non-linear curve fit function that was used for the measurement of the $IC_{50}$ is as follows:

$$y = A1 + (A2 - A1)/(1 + 10^{((LOGx0-x)*p)})$$

(where A1 = bottom asymptote, A2 = top asymptote, LOGx0 = centre and p = hill slope).

Acridine orange/ethidium bromide (AO/EB) fluorescence-based dual staining method was used to evaluate the apoptotic cell death induced by the compounds [59]. AO can go inside the intact cell membrane of a living cell and stain it green; whereas, EB can only color the apoptotic cells red/orange. After 24 h of treatment with the compounds, the cells were stained with equimolar solution of AO/EB (sigma, St. Louis, MO; 1 μL of the mixture containing 100 μg/mL of AO and EB each) and kept in dark for 3 min by covering with cover slip. Viable and apoptotic cells can be recognized as green and red/orange when observed under high-resolution microscope [60]. About 1000 cells were photographed under fluorescence microscope and the percentage of apoptosis has been determined.

### 2.7. Molecular Docking Simulation

Molecular docking is a theoretical approach that can be effectively used to predict the possible interactions of a target protein with molecules/drugs/compounds under investigation in order to dissect their possible mode of action [61]. In the present study, the molecular docking simulation was carried out for compounds **1** and **2** with the antiapoptotic cancer target proteins, viz., BCL-2 (PDB ID = 2O22) and BCL-XL (PDB ID = 2YXJ) using molegro virtual docker (MVD 2010.4.0) software [62]. The docking parameters were

run using standard parameters [63]. After molecular docking simulation; the analysis of the protein compounds was performed using chimera (https://www.cgl.ucsf.edu/chimera/ (accessed on 22 February 2022)) and discovery studio visualization-BIOVIA software (https://www.3dsbiovia.com/products/collaborative-science/biovia-discovery-studio/ (accessed on 22 February 2022)) [64].

### 3. Results

#### 3.1. Syntheses and General Aspects

[Zn$_2$Cl$_2$(H$_2$O)$_2$(μ-4-AmBz)$_2$]$_n$(**1**) has been prepared by reacting zinc(II) chloride and sodium salt of 4-aminobenzoic acid in 1:1 molar ratio at room temperature in water. [ZnCl$_2$(μ-3-AmPy)$_2$]$_n$ (**2**) has been synthesized from the reaction between zinc chloride and 3-aminopyridine in a 1:2 molar ratio under similar conditions in methanol. Both compounds are fairly soluble in water and in common organic solvents. The crystal structure of compound **2** has been re-determined with slightly different unit cell parameters with an improved goodness of fit (S) value (Table S1) from those of the previously reported structure [65]. We have performed a thorough structural investigation of the compound which unfolds non-covalent anion–π, π-stacking, and unusual NH$_2$(amino)···π interactions which stabilize the crystal packing (vide infra). We have also carried out computational studies to corroborate the presence of these supramolecular contacts in the crystal structure (vide infra).

#### 3.2. Crystal Structure Analysis

Figure 1 represents the molecular structure of compound **1**. Table 2 contains the bond lengths and the bond angles around the central Zn(II) metal centers. Compound **1** crystallizes in the monoclinic *P*2/*c* space group. The polymeric chain of compound **1** contains two different Zn(II) centers (Zn1 and Zn2) with different coordination environments. The Zn1 center is tetra-coordinated with two coordinated water (O1W and O1W') and two O-atoms (O1 and O1') of the two bridging *4-AmBz* ligands. Similarly, the Zn2 center is also tetra-coordinated with two Cl and two N-atoms (N10 and N10') of –NH$_2$ groups of the two bridging *4-AmBz* moieties. Both Zn1 and Zn2 centers have nearly tetrahedral geometries with the corresponding bond angles ranging from 90.2 to 108.2°. The average metal-ligand bond lengths are comparable to the previously reported Zn(II) compounds [66]. The Zn1–Zn1, Zn1–Zn2, and Zn2–Zn2 distances in the molecular structure of compound **1** are found to be 16.323, 9.139, and 16.323 Å, respectively.

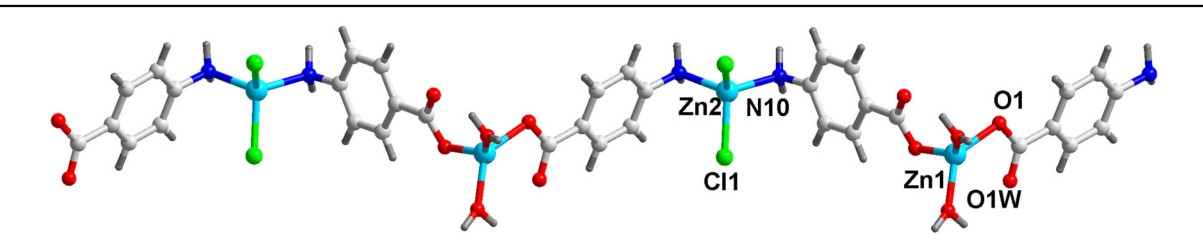

**Figure 1.** Molecular structure of [Zn$_2$Cl$_2$(H$_2$O)$_2$(μ-4-AmBz)$_2$]$_n$ (**1**).

Intramolecular O–H···O hydrogen bonding interactions are observed in the polymeric chain of compound **1** which stabilize the crystal structure (Figure S1). Uncoordinated O3 atoms of the bridging 4-AmBz and –OH (O1WH1WA) moiety of the coordinated water molecule are involved in O–H···O hydrogen bonding interactions having O1W–H1WA···O3 distance of 2.63 Å (Table 3).

**Table 2.** Selected bond lengths (Å) and bond angles (deg.) of Zn(II) centers in **1** and **2**.

| Bond | d (Å) | Angle | Degree (°) |
|---|---|---|---|
| **1** | | | |
| Zn1–O1 | 1.977(5) | O1–Zn1–O1W | 98.03(22) |
| Zn1–O1W | 1.966(6) | O1–Zn1–O1 | 103.55(20) |
| Zn2–N10 | 2.067(6) | O1–Zn1–O1W | 133.00(22) |
| Zn2–Cl1 | 2.242(2) | O1W–Zn1–O1W | 96.79(23) |
| | | O1–Zn1–O1W | 98.03(22) |
| | | N10–Zn2–Cl1 | 107.82(17) |
| | | N10–Zn2–N10 | 114.96(24) |
| | | N10–Zn2–Cl1 | 108.12(18) |
| | | Cl1–Zn2–Cl1 | 109.95(67) |
| | | N10–Zn2–Cl1 | 107.82(17) |
| **2** | | | |
| Zn1–Cl1 | 2.231(1) | Cl1–Zn1–Cl2 | 119.89(50) |
| Zn1–Cl2 | 2.236(1) | Cl1–Zn1–N1 | 109.68(14) |
| Zn1–N1 | 2.025(4) | Cl1–Zn1–N3 | 102.42(12) |
| Zn1–N3 | 2.087(6) | Cl2–Zn1–N1 | 109.37(14) |
| | | Cl2–Zn1–N3 | 103.06(13) |
| | | N1–Zn1–N3 | 112.01(21) |

**Table 3.** Selected parameters for hydrogen bonding interactions in compounds **1** and **2**.

| D–H···A | d(D–H) | d(H···A) | d(D–A) | <(DHA) |
|---|---|---|---|---|
| Compound **1** | | | | |
| C5–H5···O3 | 0.95 | 2.63 | 2.903(9) | 96.4 |
| C9–H9···O1 | 0.95 | 2.46 | 2.777(8) | 98.8 |
| C5–H5···O1 | 0.95 | 2.73 | 3.505(1) | 138.9 |
| C5–H5···O3 | 0.95 | 2.81 | 3.628(9) | 140.9 |
| C6–H6···N10 | 0.95 | 2.63 | 2.458(1) | 68.8 |
| C8–H8···N10 | 0.95 | 2.60 | 2.438(9) | 69.3 |
| O1W–H1WA···O3#1 | 0.87 | 2.63 | 3.091(7) | 114.0 |
| O1W–H1WB···O1#2 | 0.87 | 2.10 | 2.714(7) | 126.2 |
| Compound **2** | | | | |
| C2–H2···N3 | 0.95 | 2.60 | 2.445(6) | 69.9 |
| C4–H4···N3 | 0.95 | 2.64 | 2.455(6) | 68.2 |
| C6–H6···Cl1 | 0.95 | 3.00 | 3.591(4) | 121.5 |
| C6–H6···Cl2 | 0.95 | 3.03 | 3.693(5) | 127.5 |
| C4–H4···Cl1 | 0.95 | 2.82 | 3.598(5) | 139.5 |

#1 2–X, 2–Y, 1–Z; #2 2–X, 1+Y, 3/2–Z.

In the crystallographic *ac* plane, the neighboring 1D chains of compound **1** are interconnected through C–H···O and C–H···Cl hydrogen bonding and Type I Cl···Cl interactions to form the layered assembly [67] (Figure 2b). C–H···O hydrogen bonding interactions are observed involving the–CH moiety (–C5H5) of *4-AmBz* and the coordinated O1 and uncoordinated O3 atoms of the bridging *4-AmBz* having C5–H5···O1 and C5–H5···O3 distances of 2.73 and 2.84 Å, respectively. Cl1 ion and –CH moieties are involved in C–H···Cl interactions with the C8–H8···Cl1 and C9–H9···Cl1 distances of 3.05 and 3.02 Å, respectively.

Moreover, Cl···Cl interaction is also observed involving the Cl atoms (Cl1) having Cl1···Cl1 separation of 3.68 Å (Figure 2a). Usually, such intermolecular C–X1···X2–C contacts (X = F, Cl, Br, I) can be classified into two types depending on the corresponding angles, θ1 = ∠C–X1···X2 and θ2 = ∠X1···X2–C.

The interactions θ1 = θ2 are called Type I; whereas θ1 ≠ θ2 belongs to Type II contacts [68]. The Cl···Cl interactions observed in the crystal packing of **1** can be considered Type I with the corresponding angles of 127.1°. Li et al. have explored similar Cl···Cl interactions in a coordination polymer of Mn(II), viz., [Mn(tcpa)₂(bipy)]ₙ (where

Htcpa = 3,5,6-trichloropyridine-2-oxyacetic acid and bipy = 2,2′-bipyridine) [69]. We have further used computational tools to support the presence of the interactions (vide infra).

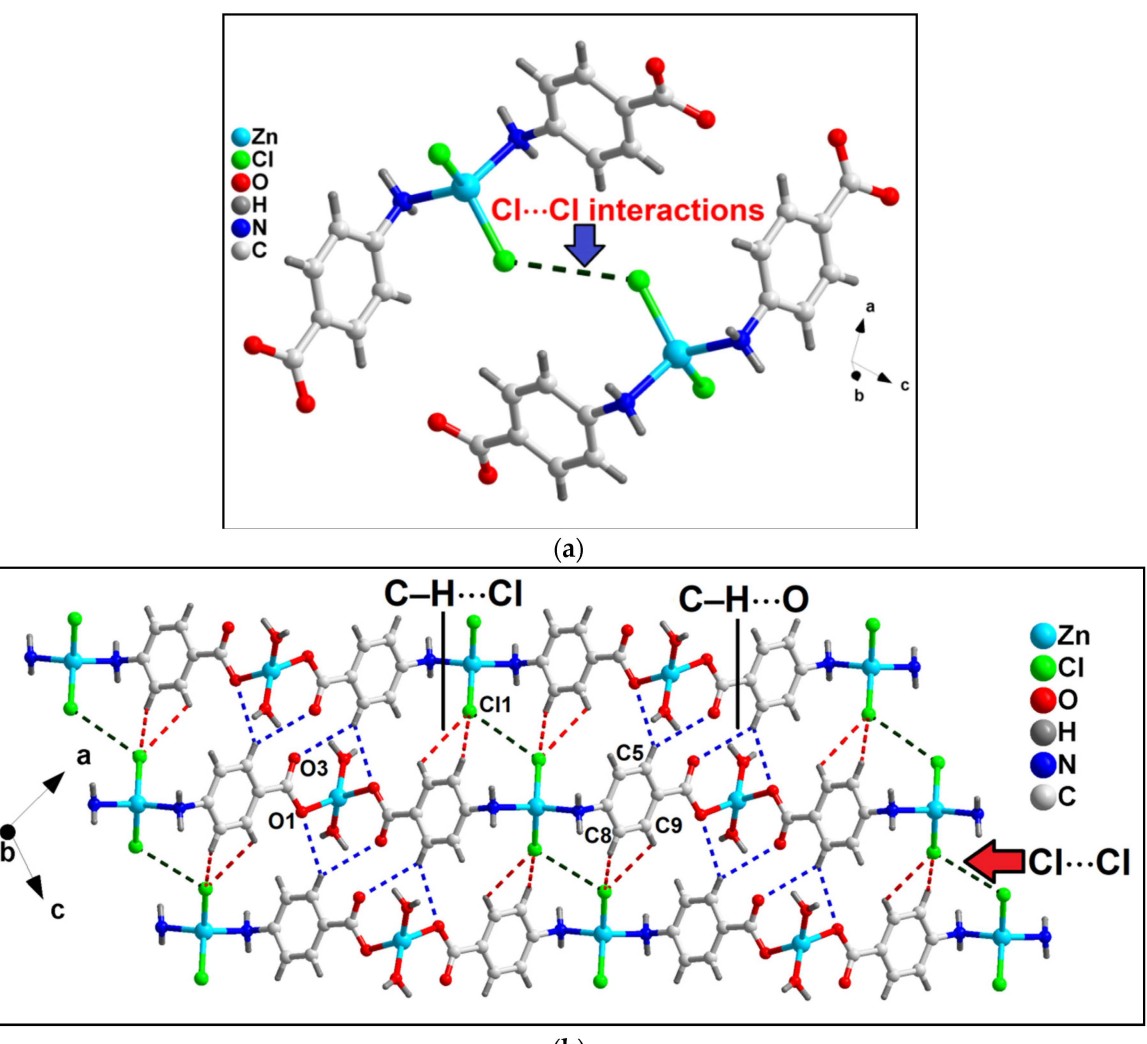

**Figure 2.** (**a**) Unconventional Type I Cl⋯Cl interactions observed in compound **1**; (**b**) 2D network architecture of compound **1** along the crystallographic *ac* plane aided by intermolecular C–H⋯O hydrogen bonding and Cl⋯Cl interactions.

Unusual NH$_2$(*amino*)⋯π, π–π and anion⋯π interactions involving the aromatic ring of coordinated *4-AmBz* ligand also stabilize the crystal structure of compound **1** (Figure 3a). NH$_2$(*amino*)⋯π interactions are observed having N⋯Cg (Cg is the ring centroid defined by the atoms C4-C9) distance of 3.67 Å. Anion⋯π contacts are observed involving the uncoordinated O3 atom of bridging *4-AmBz* moiety and the aromatic ring of *4-AmBz* with the O3⋯Cg distance of 3.68 Å (where Cg is the ring centroid defined by the atoms C4-C9). The angle of 95.2° involving O3, Cg, and the aromatic plane reveals the strong nature of the interaction.

In addition, π–stacking contacts are also observed involving the neighboring aromatic *4-AmBz* moieties with the neighboring C-C (C4–C7) separation of 3.58 Å. The dihedral angle between the aromatic rings is found to be 0.03°; whereas the slipped angle (angle between the ring normal and the vector joining the ring centroids) is found to be 17.3°. The perpendicular distance between the two aromatic rings is found to be 3.87 Å. These NH$_2$(*amino*)⋯π, anion⋯π, and π-π interactions along with the O–H⋯O and N–H⋯Cl hydrogen bonding interactions stabilize the 2D architecture of the compounds along the

crystallographic *ab* plane (Figure 3b). O–H⋯O interactions are observed involving the O1 atom of the bridging *4-AmBz* and –OH moiety (–O1WH1WB) of the coordinated water molecule having O1W–H1WB⋯O1 distance of 2.11 Å. In addition, N–H⋯Cl interactions are also observed between the –NH fragments (–N10H10B and –N10H10A) and Cl1 moiety having N10–H10B⋯Cl1 and N10–H10A⋯Cl1 distances of 2.69 and 2.86 Å, respectively.

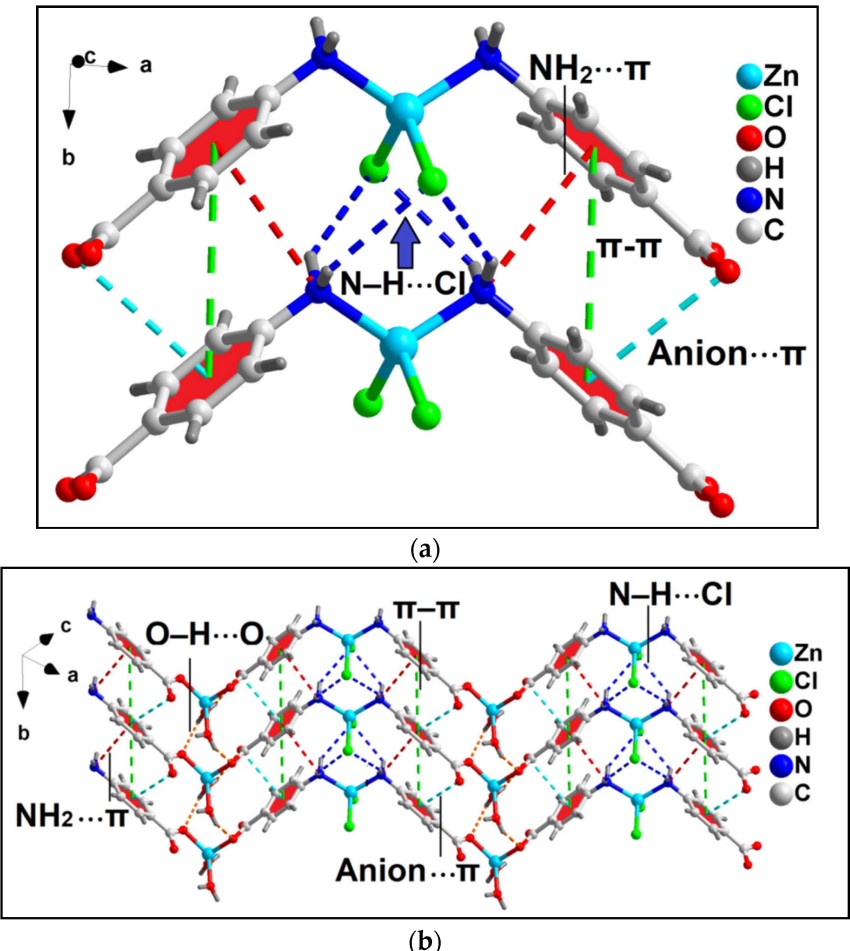

(**a**)

(**b**)

**Figure 3.** (**a**) Model dimeric assembly in the crystal structure of the polymer **1** involving anion⋯π, NH₂(*amino*)⋯π, π–π, and N–H⋯Cl interactions; (**b**) 2D network architecture of compound **1** along the crystallographic *ab* plane.

The molecular structure of compound **2** has been depicted in Figure 4. Table 2 contains the bond lengths and bond angles around the central Zn(II) center. Compound **2** crystallizes in the monoclinic *Cc* space group. Compound **2** contains one Zn(II) center, two bridging *3-AmPy* ligands, and two Cl atoms. The Zn(II) center in polymer **2** has a nearly tetrahedral coordination geometry which is formed by two Cl atoms and two N atoms of *3-AmPy*, having bond angles in the range of 102.01 to 119.8°. Zn–Cl bond lengths are in the range from 2.231 to 2.236 Å; whereas the Zn–N bond lengths are in the range of 2.025 to 2.086 Å (see Table 2) [70,71]. The adjacent Zn(II) centers of compound 2 are separated by a distance of 6.181 Å.

The presence of carboxyl, pyridine N-atom, amino groups, and aromatic rings in compounds **1** and **2** may facilitate hydrogen bond donors, acceptors, and hydrophobic regions during the compound's interaction with target proteins. H-bond donors and acceptors are the key pharmacophore features in anticancer active molecules/complexes as they are effectively involved in H-bonding interactions with the active sites of antiapoptotic proteins; thereby inhibiting the activities of antiapoptotic proteins [72]. Hydrophobic regions present in a molecule can efficiently inhibit the polymerization and repair of DNA

of cancerous cells [73]. Therefore, we have performed molecular docking studies of the compounds (*vide infra*) to investigate the possible interactions of the compounds with antiapoptotic target proteins.

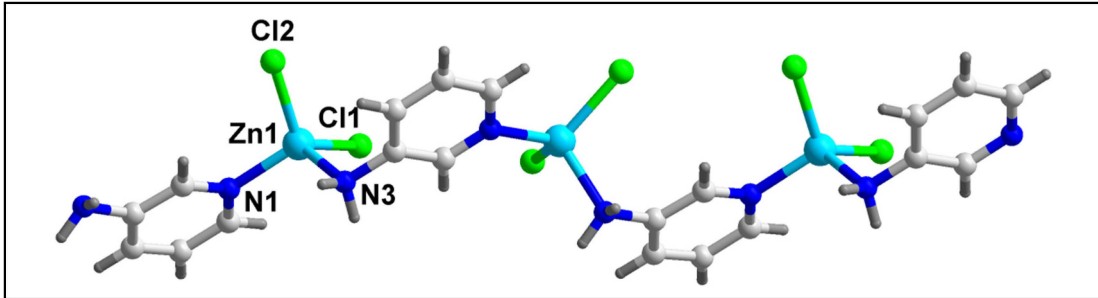

**Figure 4.** Molecular structure of [ZnCl$_2$(μ-3-AmPy)$_2$]$_n$ (**2**).

In compound **2**, intramolecular C–H···Cl hydrogen bonding interactions are observed which stabilize the 1D polymeric chain along the crystallographic *a* axis (Figure S2). Intramolecular C–H···Cl contact is observed between –CH (–C$_6$H$_6$) moiety of *3-AmPy* and Cl atom (Cl1) having C6–H6···Cl1 distance of 2.99 Å.

Anion···π, NH$_2$···π, and π–stacking interactions in addition to C–H···Cl and N–H···Cl contacts play decisive roles in the stability of the 2D architecture along the *ac* plane (Figures 5 and 6). The details of the C–H···Cl and N–H···Cl interactions observed in the 2D assembly have been tabulated in Table 3. The coordinated Cl1 and Cl2 atoms are involved in anion···π interactions with C5 and C6 atoms of bridging *3-AmPy* moiety having C5···Cl2 and C6···Cl1 distances of 3.67 and 3.53 Å, respectively. NH$_2$(*amino*)···π interactions are observed in the layered assembly of the compound with NH$_2$(*amino*)···C5 separation of 3.56 Å. In addition, π–π interactions are also observed between the aromatic rings of neighboring *3-AMpy* with the nearest C-C (C3–C6) separation of 3.60 Å. For this π-stacking interaction, dihedral and slipped angles are found to be 0.04° and 19.1°, respectively. The perpendicular distance between the two aromatic rings is found to be 3.89 Å. These interactions (shown in the model dimeric assembly; Figure 5a) have been further studied theoretically (vide infra). Similar supramolecular contacts have been obtained in another model dimeric assembly (Figure 5b) having minor differences in separation distances (C4–H4···Cl2 = 2.86 Å; N3–H3A···Cl2 = 2.50 Å; N3–H3B···Cl1 = 2.69 Å; C5···Cl1 = 3.63 Å; C6···Cl2 distances of 3.62 Å; C3–C6 = 3.60 Å) (Figure 6).

In the crystallographic *ab* plane, the 2D network is stabilized by C–H···Cl interactions (C6–H6···Cl2 = 3.04 Å) (Figure 7).

### 3.3. Spectral Studies

3.3.1. FT-IR Spectroscopy

The FT-IR spectra of compounds **1** and **2** have been recorded in the region 4000–500 cm$^{-1}$ (Figure S3). The broad absorption peak in the 3410–3450 cm$^{-1}$ region in the FT-IR spectrum for compound **1** can be attributed to the $\nu$(O–H) stretching vibrations of coordinated water molecules [74–76]. The absorption peaks of [$\nu_{as}$(OCO)] and [$\nu_s$(OCO)] bands for the carboxylate moiety of *4-AmBz* in **1**, appear at 1621 and 1388 cm$^{-1}$, respectively. The difference between the asymmetric and symmetric stretching vibrations is found to be greater than 200 cm$^{-1}$ (233 cm$^{-1}$) which supports the monodentate coordination of the carboxylate groups to Zn(II) centers in compound **1** [77]. However, the [$\nu_{as}$(NH$_2$)] and [$\nu_s$(NH$_2$)] bands of the amino groups appear at 3222, 3118, and 3107, 3228 cm$^{-1}$, respectively, for compounds **1** and **2**. These values are significantly lower than those of the free ligands which corroborates the coordination of the amino groups to the Zn(II) centers in the compounds [78].

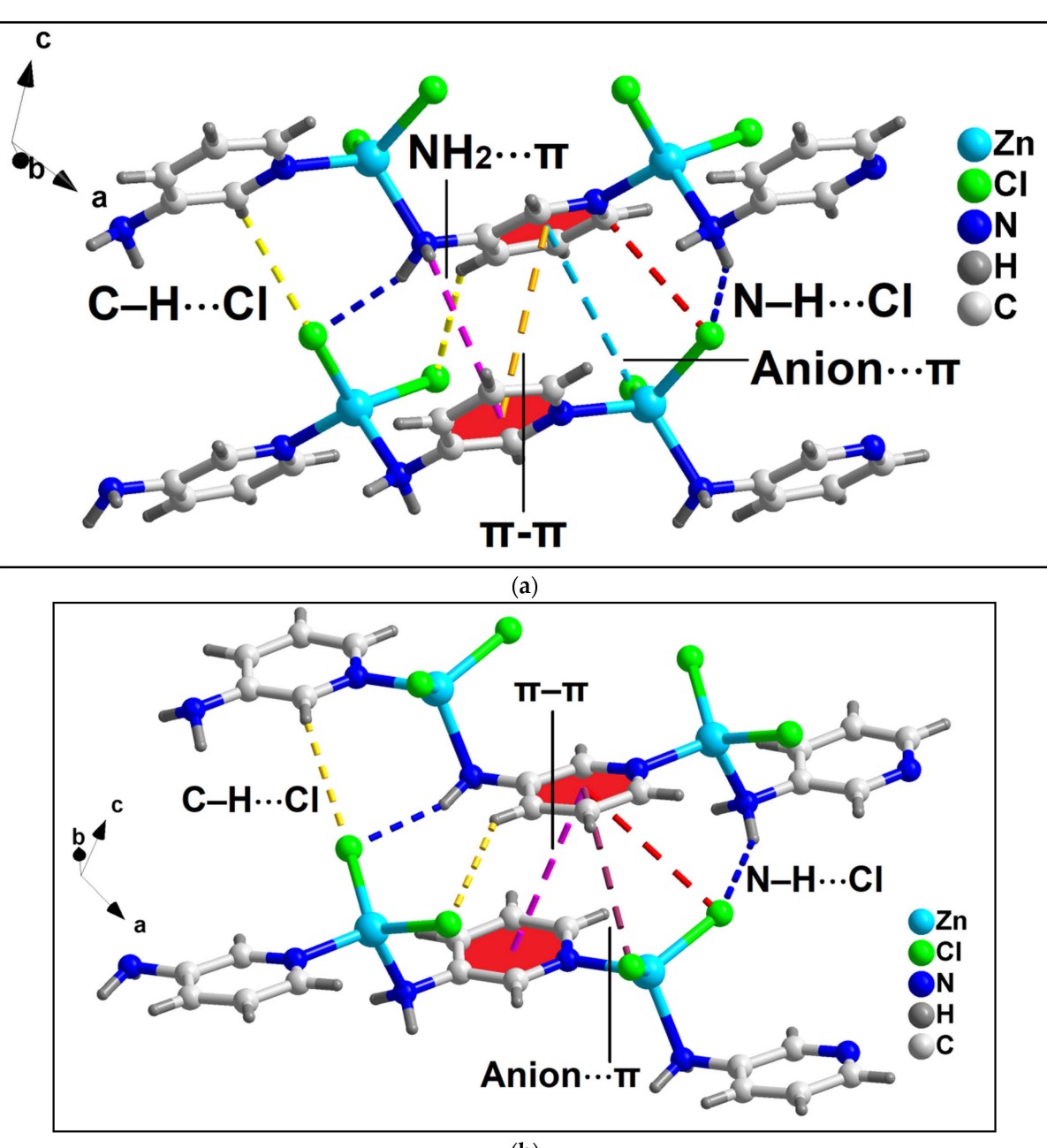

**Figure 5.** Two partial views (**a**,**b**) of the layered assembly of compound **2** along the crystallographic
*ac* plane aided by anion$\cdots \pi$, NH$_2$(*amino*)$\cdots \pi$, $\pi$–$\pi$, N–H$\cdots$Cl, and C–H$\cdots$Cl interactions.

### 3.3.2. Electronic Spectroscopy

Figures S4 and S5 depict the electronic spectra of compounds **1** and **2**, respectively, in
both the solid as well as in aqueous phases The diamagnetic Zn(II) coordination polymers
do not show absorption bands in the visible region [79]. However, peaks due to the $\pi$-$\pi$*
transitions of the aromatic ligands are observed in the UV region.

### 3.3.3. $^1$H-NMR Spectroscopy

Figures S6 and S7 depict the $^1$H-NMR spectra of compounds **1** and **2**, respectively,
in the DMSO-*d6* solvent. For compound **1**, the proton signals at 7.93 and 7.56 ppm can
be attributed to C(2)H and C(3)H protons of the benzoate ring of *4-AmBz* moiety [80]. In
compound **2**, the coordinated 3-AMpy gives proton signals at 7.68, 7.91, 7.02, 7.10 ppm due
to the C(1)H, C(2)H, C(3)H, and C(4)H protons (Figure S7), respectively [81]. The absence

of signals at >8 ppm in the spectrum of **1** supports the deprotonation of the carboxyl groups; consistent with the non-protonated *4-AmBz* present in the crystal structure [82]. The presence of coordinated water molecules in **1** can be corroborated by the presence of a signal at 3.32 ppm [83]. The signals for the amine groups in aromatic ligands may have merged with the signal of DMSO-*d6* in the NMR spectra of the compounds [84]. Therefore, the ¹H-NMR data supports the stability of the polymers in the solution phase [85].

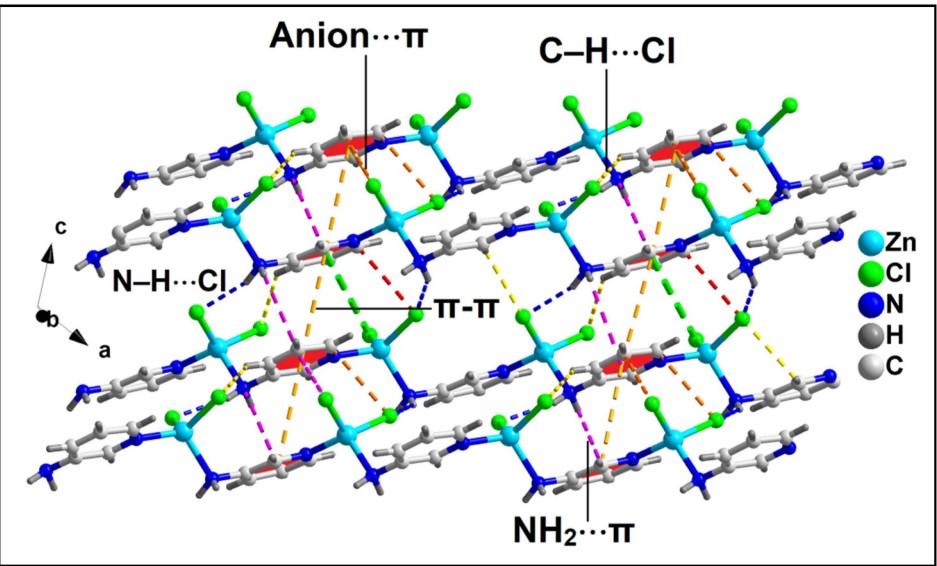

**Figure 6.** Layered assembly of compound **2** along the crystallographic *ac* plane involving anion···π, NH₂···π, and π–π interactions along with C–H···Cl and N–H···Cl hydrogen bonding interactions.

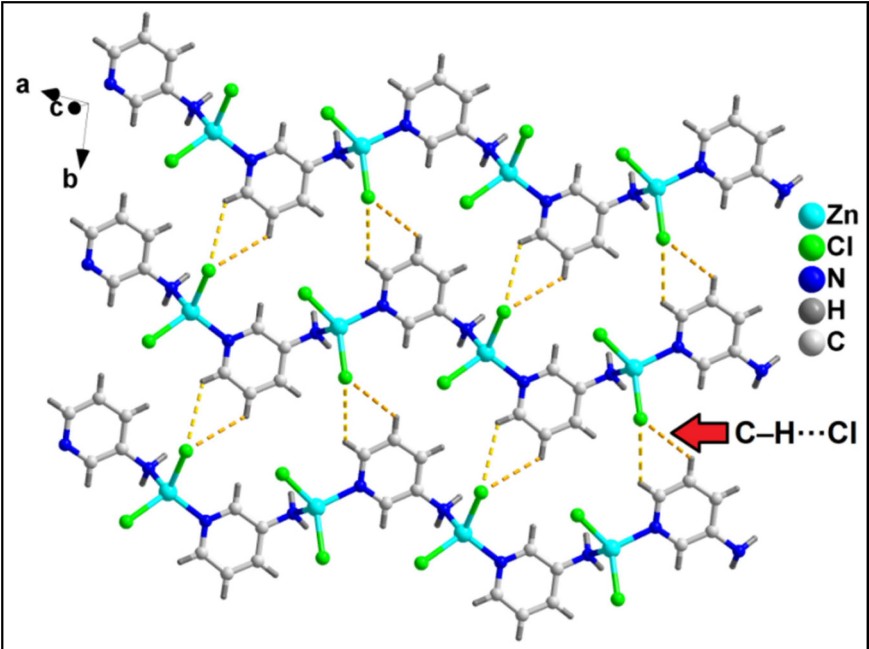

**Figure 7.** Layered assembly of compound **2** along the crystallographic *ab* plane involving C–H···Cl hydrogen bonding interactions.

### 3.4. Thermogravimetric Analysis

The thermogravimetric analysis of compounds **1** and **2** have been carried out in the temperature range of 25–800 °C under the $N_2$ atmosphere at the heating rate of 10 °C/min (Figure S8). For compound **1**; in the temperature range of 70–170 °C, two coordinated

water molecules undergo decomposition (obs. = 4.9%; calcd. = 6.2%) [86]. In the temperature range of 172–254 °C, one *4-AmBz* moiety undergoes decomposition (obs. = 28.4%; calcd. = 26.69%) [87]. Two coordinated Cl moieties are decomposed at 255–410 °C with the observed weight loss of 14.2% (calcd. = 13.92%) [88]. The compound further decomposes in an unidentified manner beyond that temperature. Compound **2** undergoes single-step decomposition in the temperature range 350–515 °C with the loss of one coordinated *3-AmPy* and two coordinated Cl moieties (obs. = 68.4%; calcd. = 70.2%) [89].

*3.5. Theoretical Studies*

As commented above, among the myriad of non-covalent interactions, the aromatic rings participate in both anion-$\pi$ (involving either the chlorido or carboxylato ligands) and $NH_2\cdots\pi$ interactions involving the coordinated amino groups. Both interactions are opposite in nature since in the former, the aromatic ring is accepting charge and in the latter is donating charge. Therefore, the aromatic ring is acting as both an electron donor and acceptor simultaneously. To shed light on this unexpected behavior, we have first computed the molecular electrostatic potential surfaces of simplified models of polymers **1** and **2** (Figure 8). The molecular electrostatic potential maximum is located at the H-atoms of the amino groups (+62 kcal/mol in **1** and +65 kcal/mol in **2**). These significant molecular electrostatic potential values are due to the enhanced acidity of the –$NH_2$ protons due to the coordination of the amino group to the $Zn^{2+}$ ion. The molecular electrostatic potential minimum is located at the chlorido ligands, as expected (−47 kcal/mol in **1** and −54 kcal/mol in **2**). These results agree with the abundance of N–H$\cdots$Cl interactions in the solid state of both compounds. In addition, more interesting are the MEP values over the center of the coordinated *4-AmBz* and *3-AmPy* ligands (Figure 8). The values are positive on one side (+9 and +21 kcal/mol in **1** and **2**, respectively), and negative over the opposite side (−15 and −7 kcal/mol in **1** and **2**, respectively), thus revealing the duality of these coordinated ligands and explaining the simultaneous formation of $NH_2\cdots\pi$ and anion–$\pi$ interactions.

We have used the (QTAIM) and (NCI plot) index analyses in an extended dimeric model of the layered structure commented above (Figure 3b), that is stabilized by H-bonds, $\pi$–stacking, anion–$\pi$, and $NH_2\cdots\pi$ interactions. The NCI plot index is an intuitive visualization index that facilitates the representation of non-covalent interactions in real space, showing which molecular regions interconnect.

The analysis of the dimer of compound **1** reveals the presence of a green and extended isosurface located between both monomers, thus confirming the strong complementarity of the polymeric chains of the compound (Figure 9). For all HBs, a small reduced density gradient (RDG) isosurface appears coincident with the location of the bond CPs (critical points). We have assessed the formation energies of the H-bonds by using the value of Vr (potential energy density) at the bond CPs and the equation proposed in the literature (E = 0.5*Vr) [90]. The anion–$\pi$ interaction is characterized by a bond critical point (CP, red sphere) and bond path (orange line) connecting the O-atom of the carboxylato group to one C-atom of the ring. Moreover, the $NH_2\cdots\pi$ is also confirmed by the QTAIM analysis, revealing a bond CP and bond path connecting the H-atom of the amino group to one C-atom of the ring. Both contacts are characterized by green RDG (reduced density gradient) isosurfaces situated between the $\pi$-cloud of the ligand and both the amino and carboxylato groups.

Interestingly, there is also a network of H-bonds that is energetically significant with a total H-bond interaction energy of −18.8 kcal/mol calculated using Vr values at the bond CPs. It can be observed that the chlorido ligands interact with the $NH_2$ groups establishing four H-bonds, each one characterized by a bond CP and bond path interconnecting the Cl and H-atoms. Similarly, the coordinated water molecules of the other Zn(II) ion also establish two H-bonds (as donors) with the coordinated carboxylato groups. The formation energy of this dimeric assembly is very large (−97.5 kcal/mol) due to this intricate combination of interactions and evidencing the importance of these contacts in the crystal

packing of **1**. Using a minimalistic model of compound **1**, we have also studied the Cl···Cl contact commented above (Figure 2).

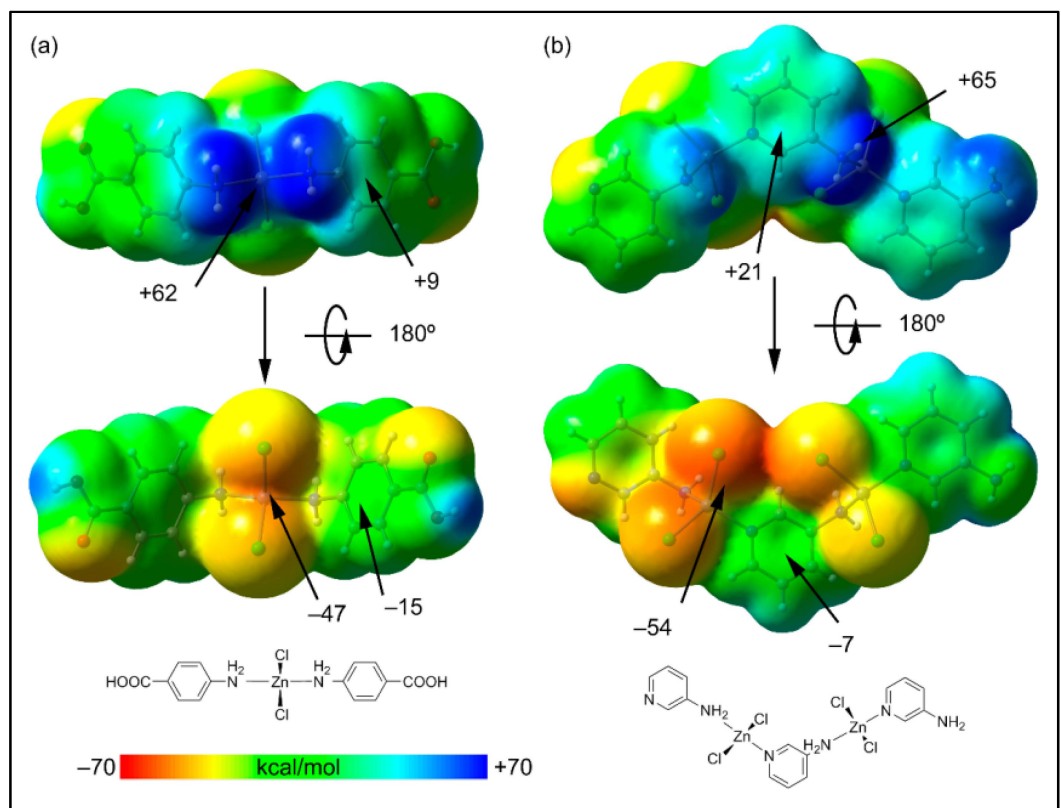

**Figure 8.** Two views of the molecular electrostatic potential surfaces of the minimalistic models of the polymers **1** (**a**) and **2** (**b**) (isosurface 0.001 a.u.). The molecular electrostatic potential energies at some points of the surfaces are given in kcal/mol. The schematic drawings of the models are indicated in the lower part of the figure.

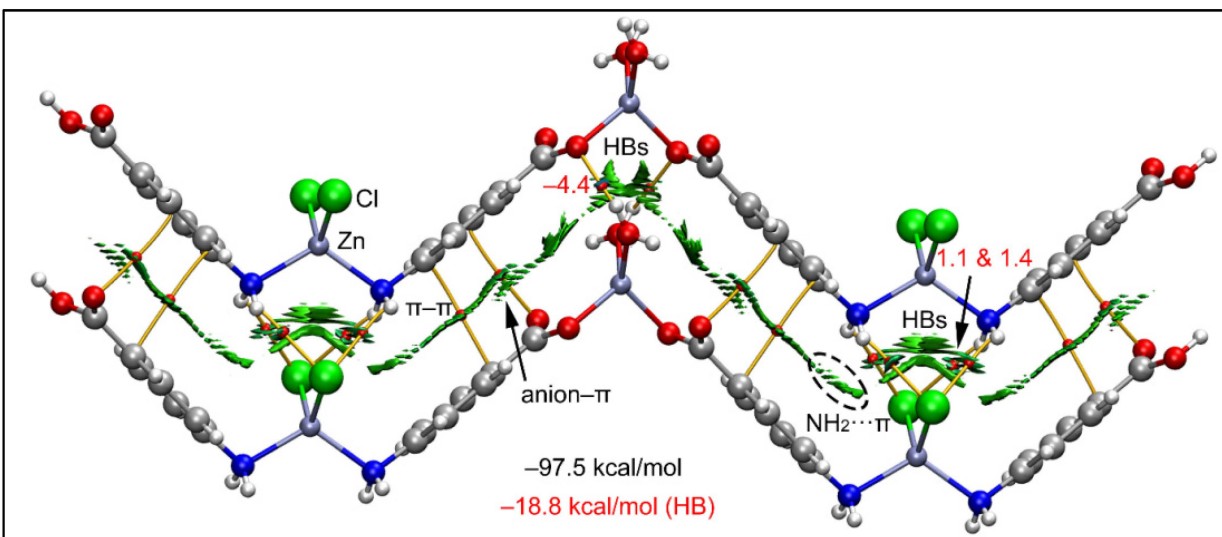

**Figure 9.** QTAIM (bond paths in orange and bond CPs in red) and RDG surface analyses of the model of compound **1**; the gradient cut-off is $\varrho = 0.04$ a.u., $s = 0.4$, and the color scale is $-0.035$ a.u. $< \varrho < 0.035$ a.u. The interaction energies are also indicated, computed at the RI-BP86-D3/def2-TZVP level of theory. Only intermolecular interactions are represented. The total contribution of the H-bonds is also indicated, estimated using the Vr energy predictor.

The joint QTAIM/NCI plot analysis is shown in Figure 10, where it can be observed that the existence of the Cl···Cl contact is corroborated by both methods (bond CP connecting the Cl-atoms and green isosurface). Moreover, the analysis also discloses the existence of other interactions, like C–H···O and C–H···Cl contacts that are also relevant for the formation of this assembly. The dimerization energy is moderately strong (−8.9 kcal/mol) where the most important contribution is the H-bonding (−7.1 kcal/mol), thus suggesting that the Cl···Cl contact is weak, as usual in Type I halogen···halogen contacts [91].

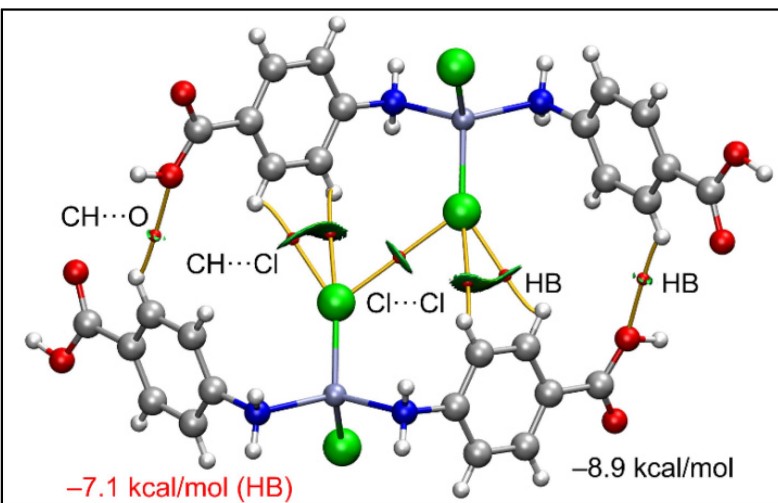

**Figure 10.** QTAIM (bond paths in orange and bond CPs in red) and RDG surfaces of the model dimeric assembly of compound **1**; the gradient cut-off is $\varrho = 0.04$ a.u., $s = 0.35$, and the color scale is $-0.04$ a.u. $< \varrho < 0.04$ a.u. Only intermolecular interactions are represented. The total contribution of the H-bonds is also indicated, estimated using the Vr energy predictor.

In compound **2**, using the model depicted in the lower part of Figure 5b, we have analyzed two dimers retrieved from its solid state. In both dimers, the aromatic ring establishes both anion-π and $NH_2···\pi$ interactions, similar to compound **1**. Both dimers are represented in Figure 11 exhibiting similar dimerization energies. The π-interactions (π–π, $NH_2···\pi$ and anion–π) are characterized by the corresponding bond CPs and bond paths interconnecting the atoms of the counterparts. All three interactions are also revealed by the NCI plot analysis and characterized by green RDG isosurfaces, confirming their attractive nature. The analysis also shows the presence of C–H···Cl and N–H···Cl H-bonding interactions that contribute −4.7 and −7.5 kcal/mol for the dimers in Figure 11a,b, respectively, thus evidencing that the π-interactions dominate the formation of the dimers. These results evidence that π-interactions are more relevant than the H-bonds in the crystal packing of compound **2**.

### 3.6. Cytotoxicity and Apoptosis Assays

Trypan blue is a polar dye that cannot penetrate through the intact cell membrane of normal cells; but can enter and color the non-viable cells due to damaged membrane integrity [92]. The percentage of dead cells can be calculated from the number of colored (stained) cells compared to the total population of cells [93]. The results of the trypan blue assay reveal that compounds **1** and **2** induced remarkable concentration-dependent cytotoxicity in DL cancer cells with negligible cytotoxicity in PBMC cells (Figures 12 and 13). The cytotoxicity assay of the metal salt ($ZnCl_2$) and the corresponding ligands (*4-AmBz*, *3-AmPy*) under the same experimental conditions reveal that they exhibit negligible cytotoxicity against DL cancer cell after 24 h of treatment. The study reveals that the significant cytotoxicity exhibited by the compounds in a cancerous cell line is a combined effect of the metal center and the ligands [94]. Comparative analysis revealed that compound **1** induces better cytotoxicity in DL cells compared to that of compound **2**. The higher positive charge of

Zn(II) center in compound **1** than in compound **2**; due to the smaller Zn-ligand bond length (2.06 Å in **1**; 2.14 Å in **2**); can be correlated to the higher cytotoxicity of compound **1** [95]. In addition, compound **1** interacts more efficiently with BCL-2 antiapoptotic protein than that of compound **2** (vide infra); which may also play a significant role in more cytotoxicity of **1**.

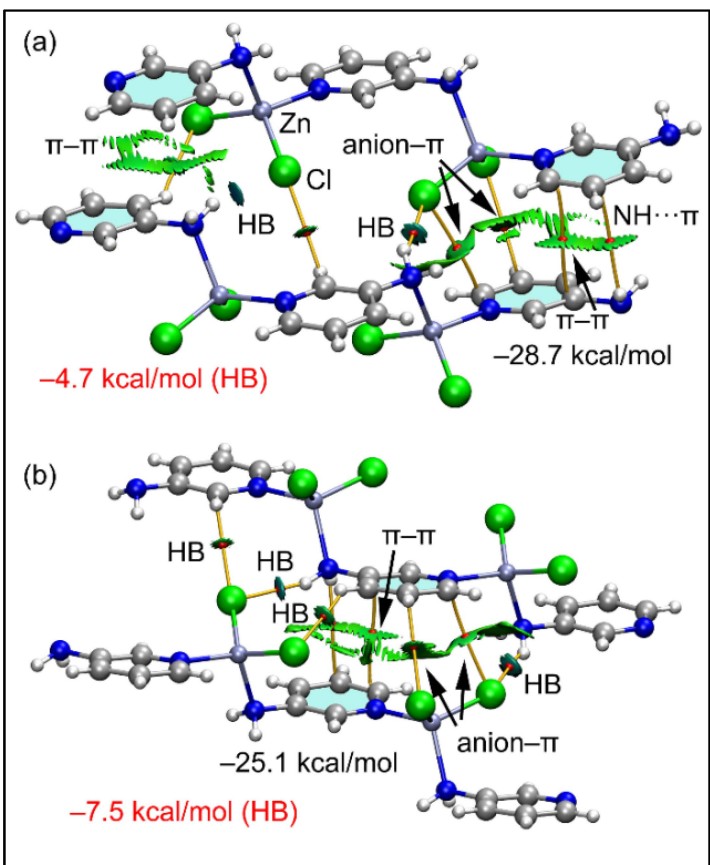

**Figure 11.** (**a**,**b**) QTAIM (bond paths in orange and bond CPs in red) and RDG surfaces of two dimeric models of compound **2**. The gradient cut-off is $\varrho$ = 0.04 a.u., $s$ = 0.35, and the color scale is −0.04 a.u. < $\varrho$ < 0.04 a.u. Only intermolecular interactions are represented. The total contribution of the H-bonds is also indicated, estimated using the Vr energy predictor.

We have also carried out an AO/EB dual staining approach to investigate the apoptotic cell death induced by the compounds in DL cancer cells. AO is a vital dye that can penetrate the cell membrane of normal healthy cells and stain the nucleus as green; whereas, EB can only penetrate the damaged cell membrane of apoptotic cells and stain as red/orange [96]. The presence of red/orange fluorescence after exposure to compounds **1** and **2** indicates apoptosis-inducing abilities of the compounds (Figures 12 and 13). A comparative study revealed that compound **1** induced higher apoptosis compared to compound **2** and the result is in line with that of the cytotoxicity assay (trypan blue exclusion assay). The results obtained for the cytotoxicity and apoptosis assays have been further compared with cisplatin (Sigma Aldrich), a reference drug. There are few reports of significant antiproliferative evaluation of coordination polymers involving bridging amine-based ligands in human cancerous cells [97,98]. Li and coworkers have reported the in vitro antiproliferative evaluation of a Co(II) coordination polymer with *4-AmBz* derivatives against human liver cancer cells [99].

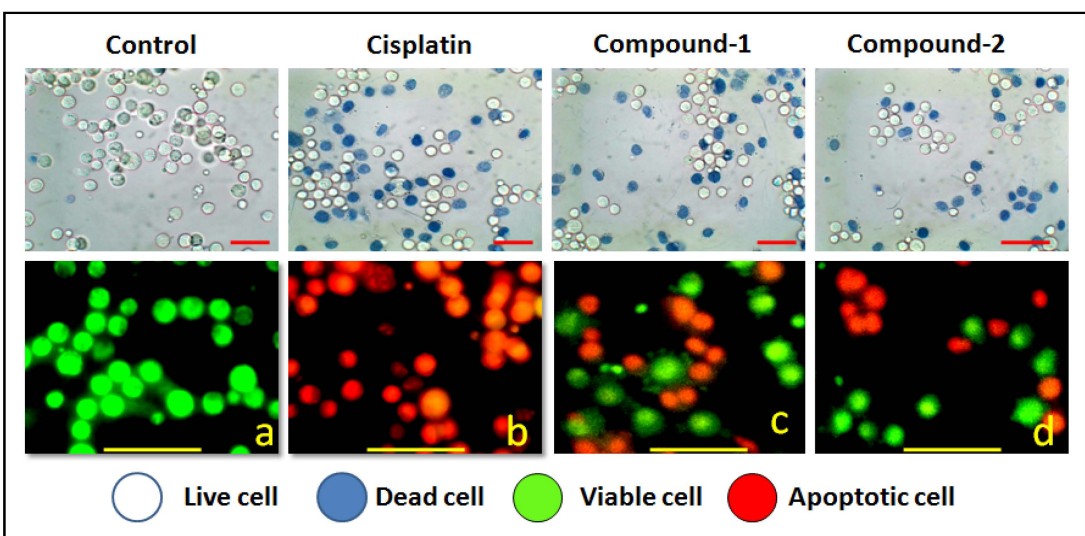

**Figure 12.** Upper panel shows the results of trypan blue assay of the compound's DL cells. Lower panel represents the morphological feature of living (green) and apoptotic (red) DL cancer cells observed under fluorescence microscope. Figure corresponds to apoptosis (450×) induced by the compounds at highest potent dose (10 μM).

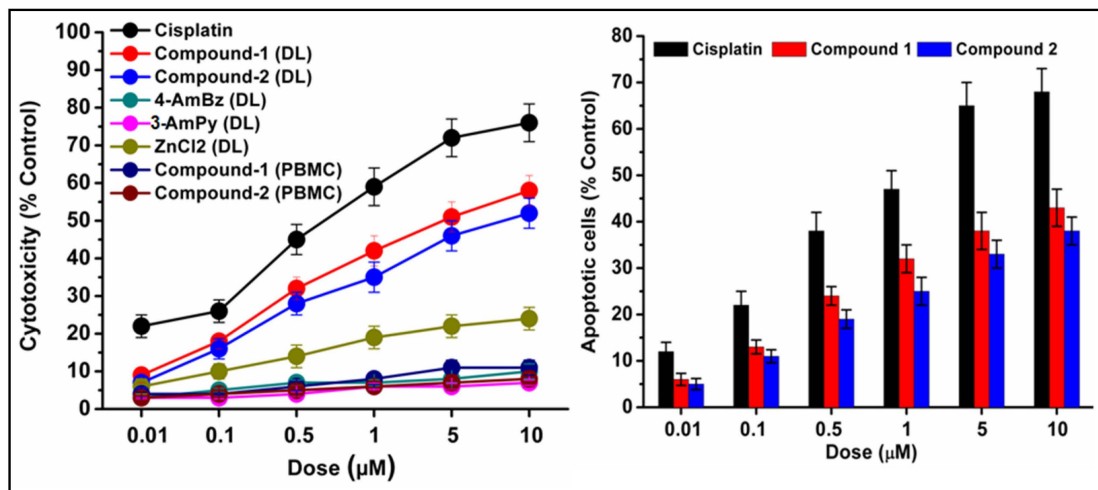

**Figure 13.** Concentration-dependent cytotoxicity (left) induced by compounds **1** and **2** in DL cells. The results of the compounds were compared with cisplatin (reference drug). Percentage apoptotic cells (right) in DL cell line after exposure to compounds **1** and **2**. Data are mean ± S.D., n = 3.

The preclinical assessment of newly synthesized drugs/compounds can be typically evaluated by the half maximum inhibitory concentration ($IC_{50}$) which can be measured by the substance's ability to inhibit 50% of biological processes [100]. The dose-response curves and the $IC_{50}$ values of the compounds have been represented in Figure 14 and Table S2, respectively. Compound **1** exhibits a lower $IC_{50}$ value (5.8 μM) than that of compound **2** (16.8 μM) in DL cells after 24 h of treatment (Figure 14). The $IC_{50}$ value of cisplatin (reference drug) ranges from 0.5–0.6 μM under the same experimental conditions. Several research groups have reported the significant anticancer activities of transition metal compounds against DL cells [38–40]. However, they have not investigated the cytotoxicity of the compounds in normal cell lines. In our study, the promising outcome of the results of the cytotoxicity assay is that the compounds induced significant concentration-dependent cytotoxicity in DL cancer cells with negligible cytotoxicity in PBMC cells. Moreover, most of the previous reports do not perform the molecular docking simulation (vide infra) to establish the probable mechanism of the synthesized compounds in the cytotoxic assay.

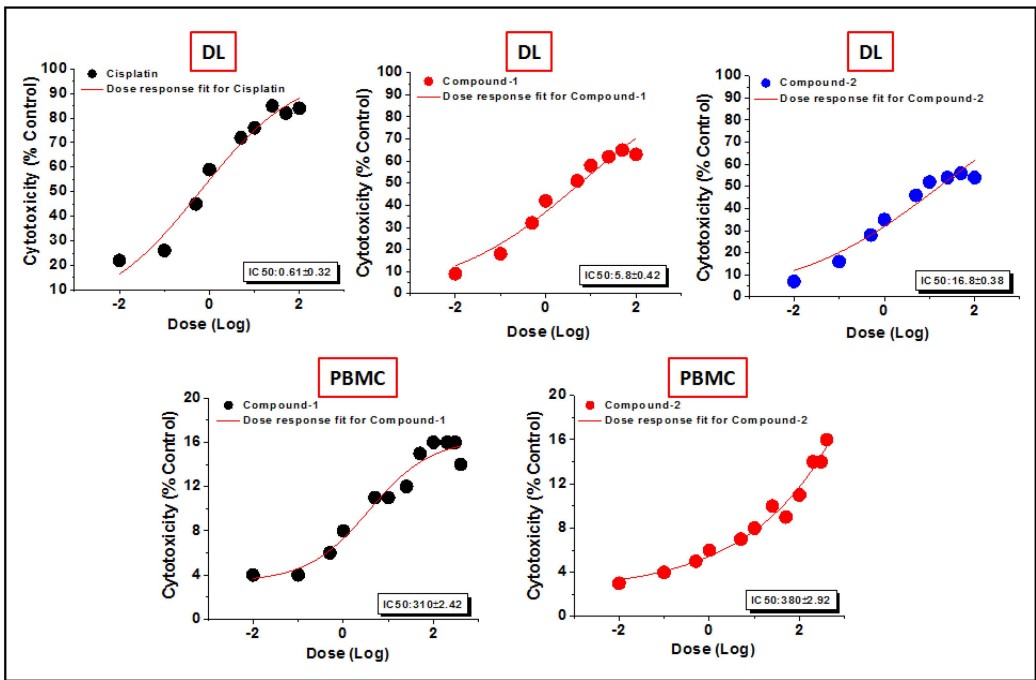

**Figure 14.** The dosage response curves of compounds **1** and **2** to determine the IC$_{50}$ values in DL and PBMC cell lines.

### 3.7. Molecular Docking Simulation

In the present study, in silico molecular docking simulations of the structures of the compounds have been performed to explore the probable interactions of the compounds with antiapoptotic BCL family proteins which are closely related to cancer progression and metastasis [101]. Molecular docking is a potential approach in computer-aided drug designing (CADD) for the virtual screening of drugs with the 3D structures of biological target proteins [102]. Molecular docking simulation was performed with the antiapoptotic target proteins [BCL-2 (PDB ID = 2O22) and BCL-XL (PDB ID = 2YXJ)] to elucidate the possible binding modes of the modeled fragments of the compounds with the active sites of the receptors. It has been well established that antiapoptotic BCL family proteins are actively involved in cancer cell progression and metastasis [103–105]. Studies on BCL family proteins have revealed their active involvement in tumor pathogenesis and therefore they can be effectively utilized as targets for developing more precisely tailored drugs/chemicals [62]. Figures 15 and 16 represent the docking structures of the compounds (modeled fragment) with the antiapoptotic target proteins which reveal significant binding affinity of the modeled fragments of the compounds with the receptor proteins. Compound **1** possesses a stronger binding affinity with anti-apoptotic protein BCL-2 than that of compound **2**, while interactions with BCL-XL protein revealed similar arrangements in both compounds. We have already mentioned in the cytotoxic assay that the comparatively higher positive charge of the Zn(II) center in compound **1** due to smaller Zn-ligand bond lengths than that of compound **2** can be correlated to the higher cytotoxicity of compound **1** [95]. The metal-ligand bond lengths and bond angles play crucial roles in the fitting of the compound in the receptor-binding pockets of the target proteins [106]. The smaller Zn-ligand bond lengths in **1** may play a crucial role in higher binding affinity with the anti-apoptotic protein BCL-2.

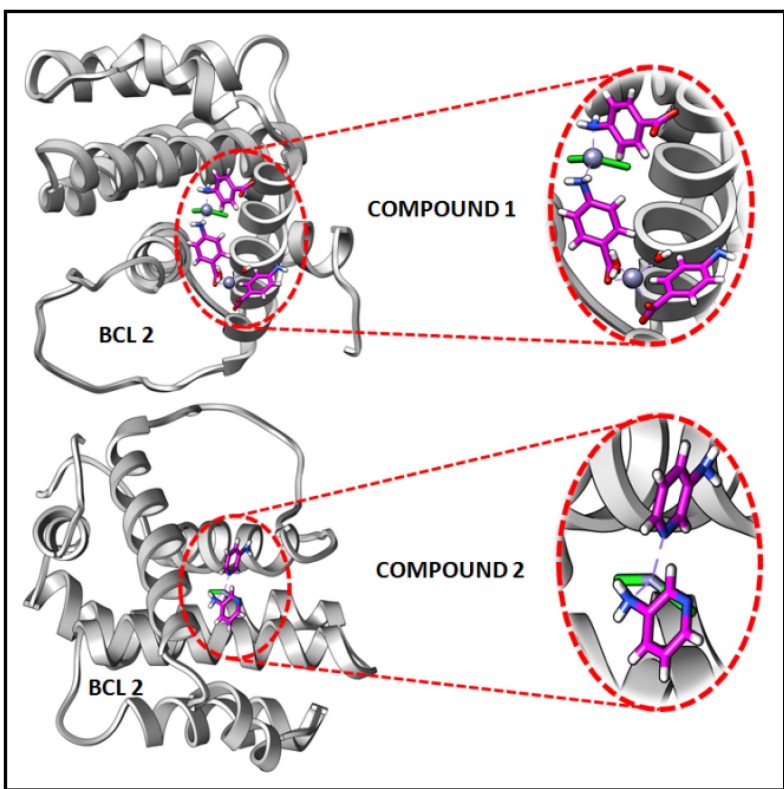

**Figure 15.** Docking results of modelled fragments of the compounds **1** and **2** with BCL-2 receptor.

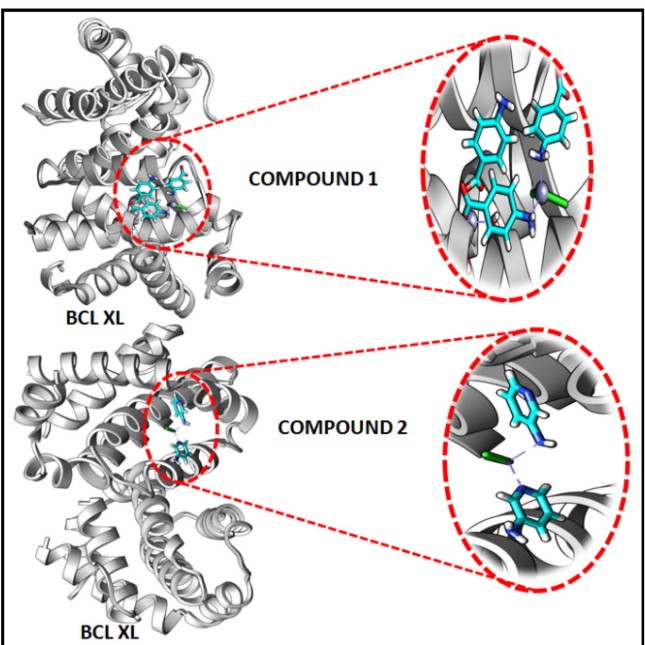

**Figure 16.** Docking results of modelled fragments of the compounds **1** and **2** with BCL-XL receptor.

Figures 17 and 18 represent the chemical interactions between the compounds and the amino acid residues with the active sites of antiapoptotic proteins. The docking score of compound **1** for BCL-2 and BCL-XL proteins were −99.92 and −86.29, respectively, whereas, for compound **2**, the scores are −59.78 and −62.59, respectively. Compound **1** interacted strongly with the active sites of BCL-2 protein having amino acid residues Tyr19, Arg95, Arg106, Gln96, Asp99, and Asp100, whereas, single interaction was observed in the active sites of BCL-XL protein having Glu129(B) residue (Figures 17 and 18). On the other

hand, compound **2** interacted with four amino acid residues, viz., Arg95, Asp99, Ser102, and Arg103 in BCL-2 protein, while single interaction was observed in BCL-XL protein with Phe105(B) (Figures 17 and 18). Thus, it can be stated that the compounds' efficient ability to induce cytotoxicity and apoptosis may be due to their effective interactions with the antiapoptotic target proteins. Konakanchi et al. have supported the experimental anticancer activities of five coordination compounds of Co(II), Ni(II), Cu(II), Zn(II), and Pd(II) involving substituted aminobenzoic acid using molecular docking studies [107].

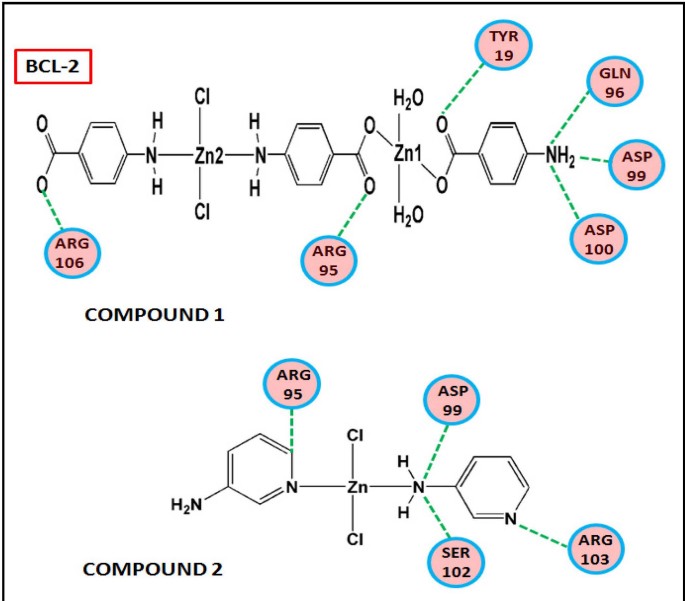

**Figure 17.** Chemical interactions of modelled fragments of the compounds with BCL-2 have been represented with compounds' atoms and interacting amino acid residues.

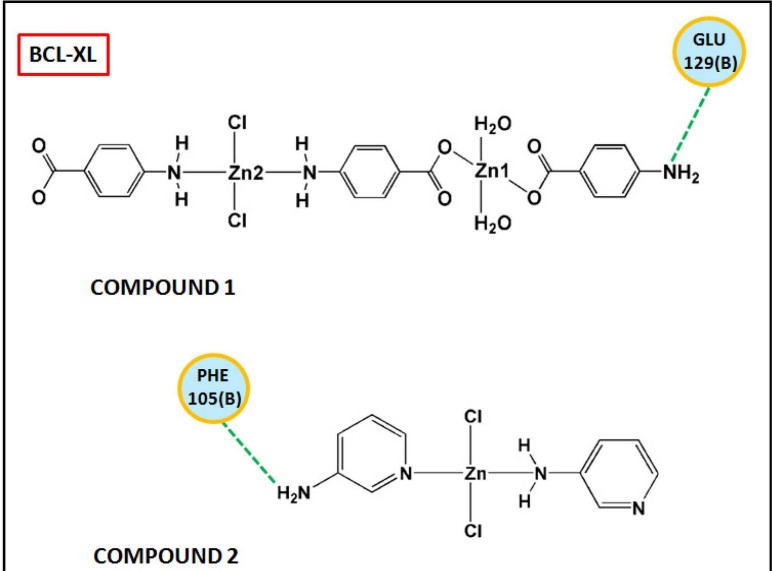

**Figure 18.** Chemical interactions of modelled fragments of the compounds with BCL-XL have been represented with compounds' atoms and interacting amino acid residues.

## 4. Conclusions

Two Zn(II) coordination polymers have been prepared at room temperature and characterized using elemental analysis, FT-IR, electronic spectroscopy, TGA, and single crystal XRD. Compounds **1** and **2** crystallize as 4-aminobenzoato and 3-aminopyridine bridged

coordination polymers of Zn(II), respectively. Non-covalent anion-π, aromatic π-stacking, and unusual NH₂(*amino*)···π contacts are observed in the crystal structures which provide rigidity to the layered assemblies. Unorthodox Type I Cl···Cl interactions also play a pivotal role in the stabilization of the layered assembly of compound **1**. The non-covalent interactions observed in the solid-state architectures of the compounds have been further analyzed theoretically using DFT calculations. Interestingly, the MEP surface analysis evidenced interesting duality in donor-acceptor topologies of the aromatic rings of coordinated *4-AmBz* and *3-AMpy* moieties in the compounds revealing the concurrent formation of unusual NH₂···π and anion–π interactions. DFT calculations, NCI plot index, and QTAIM analysis reveal that among various non-covalent interactions involved in the crystal packing of the compounds, H-bonds in compound **1** and π-interactions (NH₂···π, π-π, anion–π) in compound **2** are energetically significant. We have carried out in vitro anticancer activities of the compounds in Dalton's lymphoma (DL) malignant cancer cell line using trypan blue exclusion and apoptosis assays. The study reveals concentration-dependent cytotoxicity and apoptosis-inducing abilities of the compounds in cancerous cells having low effects in normal PBMC cells. The compounds efficiently bind with the active sites of antiapoptotic proteins which is evident from the molecular docking simulation.

**Supplementary Materials:** The following supporting information can be downloaded at: https://www.mdpi.com/article/10.3390/cryst13030382/s1, Figure S1: 1D polymeric chain of compound **1** assisted by intra-molecular O–H···O hydrogen bonding interactions; Figure S2: 1D polymeric chain of compound **2** stabilised by C–H···Cl hydrogen bonding interactions; Figure S3: FT-IR spectra of the compounds **1** and **2**.; Figure S4: (**a**) UV-Vis-NIR spectrum of **1**; (**b**) UV-Vis spectrum of **1** in water ($10^{-3}$ M); Figure S5: (**a**) UV-Vis-NIR spectrum of **2**; (**b**) UV-Vis spectrum of **2** in water ($10^{-3}$ M); Figure S6: $^1$H-NMR spectrum of compound **1** in DMSO-*d6*; Figure S7: $^1$H-NMR spectrum of compound **2** in DMSO-*d6*; Figure S8: Thermogravimetric curves of the compounds **1** and **2**; Table S1: Comparison of crystal parameters of compound **2** with the already reported compound; Table S2: IC$_{50}$ values (in μM) of compounds **1**, **2**, ligands and metal salt calculated using dose response curves.

**Author Contributions:** Conceptualization, A.F., A.K.V. and M.K.B.; methodology, A.F., A.K.V. and M.K.B.; software, A.F., R.M.G. and A.K.V.; formal analysis, A.F.; investigation, P.S., R.M.G. and D.D.; data curation, M.B.-O.; writing—original draft preparation, P.S. and M.K.B.; writing—review and editing, M.K.B.; visualization, A.F.; supervision, M.K.B.; project administration, A.F. and M.K.B.; funding acquisition, A.F. and M.K.B. All authors have read and agreed to the published version of the manuscript.

**Funding:** Financial support from ASTEC, DST, Govt. of Assam (Grant number: ASTEC/S&T/192(177)/2020-2021/43) and the Gobierno de España, MICIU/AEI (project No. PID2020-115637GB-I00 FEDER funds) is gratefully acknowledged. The authors thank IIT, Guwahati for TGA data.

**Data Availability Statement:** Not applicable.

**Conflicts of Interest:** The authors declare no conflict of interest. The funders had no role in the design of the study; in the collection, analyses, or interpretation of data; in the writing of the manuscript; or in the decision to publish the results.

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
