# Peer review of "Unconventional Dual Donor-Acceptor Topologies of Aromatic Rings in Amine-Based Polymeric Tetrahedral Zn(II) Compounds Involving Unusual Non-Covalent Contacts: Antiproliferative Evaluation and Theoretical Studies"

_crystals, doi:10.3390/cryst13030382_

Round 1
Reviewer 1 Report
This work describes two new Zn complexes, analyzed by diffraction and spectral methods. The main attention in the article is paid to the structural part, the packing of polymer chains, intra- and intermolecular interactions. And also, the work is supplemented with interesting information about cytotoxicity, anticancer activity, enriched with molecular docking. I find this work interesting and suitable for publication in a journal.
I have some comments that may improve the quality of the article.
1. Supplement the description of stacking interactions with data on the characteristic dihedral angle between the planes, perpendicular to the distance Cg(I) on the ring J, and slippage.
2. There are many shortcomings in the text due to the lack of space between words and idioms.
3. Real centroids and connections between them are better shown in the figures.
I recommend to publish this paper.
Author Response
First, we would like to thank this reviewer for his/her careful reading of the manuscript, corrections and suggestions. Our changes and "point by point" replies follow:
This work describes two new Zn complexes, analyzed by diffraction and spectral methods. The main attention in the article is paid to the structural part, the packing of polymer chains, intra- and intermolecular interactions. And also, the work is supplemented with interesting information about cytotoxicity, anticancer activity, enriched with molecular docking. I find this work interesting and suitable for publication in a journal.
I have some comments that may improve the quality of the article.
- Supplement the description of stacking interactions with data on the characteristic dihedral angle between the planes, perpendicular to the distance Cg(I) on the ring J, and slippage.
Reply: The details have been incorporated in the revised manuscript as suggested by the esteemed reviewer.
- There are many shortcomings in the text due to the lack of space between words and idioms.
Reply: We have corrected this in the revised manuscript.
- Real centroids and connections between them are better shown in the figures.
Reply: We have revised the figures accordingly.
I recommend to publish this paper.
Reviewer 2 Report
The acronyms must be explained when they first appear in the text, e.g. QTAIM, NCI, HBs etc.
Section Materials and Methods: the characteristics of the diffractometer and the working conditions must be presented here.
The authors are asked if the polymer structure of the compounds in the crystalline state still remains in solution, especially in the DMSO solution (considering the dissociation capacity of this solvent). Also, the authors must specify if apart from their analytical value, the crystallographic data are representative/significant for the structural characteristics of the compounds on which the interaction with proteins depends and, therefore, their biological activity, especially since the interaction with proteins (according to molecular docking simulation) is done through well-defined structural units (Zn2L3 for compound 1 and Zn1L2 for compound 2).
Those non-covalent interactions that appear in complexes in the crystalline state (interactions highlighted by X-ray diffraction) how relevant are they for their stability and biological activity, as the title of the manuscript suggests?
It would be useful if the bands in the IR spectra (mentioned and assigned in the text) were assigned/identified by the authors in the recorded spectra (Fig. S3), similar to the signals in the NMR spectra.
Regarding the interactions with proteins, if the carboxylate oxygen and the pyridinic nitrogen function as hydrogen bond acceptors, how do the amine groups, considering that the nitrogen atom, by the non-participating electrons, is involved in the interaction with zinc cation and in figures 17 and 18 these amine groups are represented both as donors and as acceptors of hydrogen bonds?
What is the opinion of the authors: can a single interaction with the BCL-XL protein explain the cytotoxic activity of the compounds?
Lines 414-415: The phrase "The lack of any resonances at >8 ppm in the spectrum of 1 clearly demonstrates the absence of acidic protons of carboxylate groups" should be reformulated, because the carboxylate has no proton, the carboxyl group has a proton.
If the authors highlight and explain the absence in the NMR spectrum of the signal of the proton from the carboxyl group, how do they explain the absence of the signals corresponding to the protons from the amine groups? By proton mobility?
Author Response
First, we would like to thank this reviewer for hisher careful reading of the manuscript, corrections and suggestions. The changes made are listed below:
The acronyms must be explained when they first appear in the text, e.g. QTAIM, NCI, HBs etc.
Reply: The acronyms have been explained in the revised manuscript as suggested by the esteemed reviewer.
Section Materials and Methods: the characteristics of the diffractometer and the working conditions must be presented here.
Reply: The characteristics of the diffractometer and the working conditions have been represented in section 2.1.
The authors are asked if the polymer structure of the compounds in the crystalline state still remains in solution, especially in the DMSO solution (considering the dissociation capacity of this solvent).
Reply: We have recorded the 1H-NMR spectra of the compounds in DMSO-d6 solvent which is consistent with the crystal structures of the compounds. Various research groups have explored the solution phase stability of coordination compounds by NMR spectroscopy in DMSO-d6 solvent (https://doi.org/10.3390/ijms22168704; https://doi.org/10.1002/aoc.6508; https://doi.org/10.1039/D1DT00412C). We have also reported the solution phase stability of two poly-nuclear coordination compounds using NMR spectroscopy in DMSO-d6 solvent and explored in vitro anticancer activities (New J. Chem., 2021, 45, 13040-13055; https://doi.org/10.1039/D1NJ00619C).
Also, the authors must specify if apart from their analytical value, the crystallographic data are representative/significant for the structural characteristics of the compounds on which the interaction with proteins depends and, therefore, their biological activity, especially since the interaction with proteins (according to molecular docking simulation) is done through well-defined structural units (Zn2L3 for compound 1 and Zn1L2 for compound 2).
Reply: The presence of carboxyl, pyridine N-atom, amino groups and aromatic rings in the compounds 1 and 2 facilitate as hydrogen bond donors, acceptors and hydrophobic regions during the compound’s interaction with target proteins. H-bond donors and acceptors are the key pharmaceutically active features in anticancer active molecules/complexes as they are effectively involved in H-bonding interactions with the active sites of antiapoptotic proteins; thereby inhibiting the activities of antiapoptotic proteins (https://doi.org/10.1007/s10989-020-10150-3). It has been well established that hydrophobic regions are efficient in inhibiting the polymerization and repair of DNA of cancerous cells. Therefore, we have carried out molecular docking studies of the compounds to explore the possible interactions of the compounds with antiapoptotic target proteins. However, the docking simulation cannot be correctly applied to synthetic polymers with bridging chains (10.1007/s10822-012-9621-7). Therefore; we have used the minimum repeating units of the compounds (Zn2L3 for compound 1 and Zn1L2 for compound 2) to dock with the antiapoptotic target proteins. In the revised manuscript, we have used the term ‘modelled fragment’ instead of the whole compounds in docking part.
Those non-covalent interactions that appear in complexes in the crystalline state (interactions highlighted by X-ray diffraction) how relevant are they for their stability and biological activity, as the title of the manuscript suggests?
Reply: In the crystal structure analysis section of the compounds, we unfold the presence of non-covalent anion–π, π-stacking, unusual NH2(amino)⋯π interactions, Unconventional Type I Cl⋯Cl interactions which play pivotal role in the stabilization of the compounds. These interactions observed in the crystal structures have been further studied theoretically in order to explore the energetic features of the interactions.
We have carried out the in vitro anticancer activities of the compounds to explore the potential biological applications of the compounds. Molecular docking is one of the extensively utilized Computer-Aided Drug Designing techniques to design pharmaceuticals by optimizing potential candidates targeted against some specific proteins (https://doi.org/10.1021/jm051197e; https://doi.org/10.1007/978-1-59745-177-2_19). To corroborate the promising anticancer activities of the compounds 1 and 2 considering cell cytotoxicity and apoptosis, we have further performed molecular docking studies of the compounds with antiapoptotic BCL family target cancer proteins which are directly related to cancer progression and metastasis. It has been well established that antiapoptotic proteins viz. BCL-2, BCL-XL etc. play crucial roles in apoptotic cell death of human cancer cell lines (https://doi.org/10.1016/j.bbamcr.2013.08.006). Studies on BCL family proteins have revealed the active involvement of them towards tumor pathogenesis and therefore can be effectively utilized as targets for developing more precisely tailored drug/chemicals (https://doi.org/10.1007/s10495-008-0300-z). Compounds 1 and 2 interacted with the active sites of the antiapoptotic proteins via various non-covalent interactions; thereby inhibiting their activities and resulting apoptosis.
Therefore, the title is reflecting the importance of non-covalent contacts towards stability of the compounds as well as the properties of the compounds.
It would be useful if the bands in the IR spectra (mentioned and assigned in the text) were assigned/identified by the authors in the recorded spectra (Fig. S3), similar to the signals in the NMR spectra.
Reply: We have now assigned the bands in IR spectra. The IR spectrum of compound 2 has been recorded again as the band intensity for NH2 stretching vibrations were very weak in the previous spectrum.
Regarding the interactions with proteins, if the carboxylate oxygen and the pyridinic nitrogen function as hydrogen bond acceptors, how do the amine groups, considering that the nitrogen atom, by the non-participating electrons, is involved in the interaction with zinc cation and in figures 17 and 18 these amine groups are represented both as donors and as acceptors of hydrogen bonds?
Reply: Carboxylate oxygen and the pyridinic nitrogen atoms do not contain any H-atoms; so they cannot act as hydrogen bond donors during interactions with proteins. However; the amine (NH2) can effective act as a hydrogen bond donor due to presence of two N-H bonds. Coordination of the ligand to the metal center can effectively polarize the electron cloud around the ligand (New J. Chem., 2021, 45, 13040-13055; https://doi.org/10.1039/D1NJ00619C). Amine groups of the compounds are acting both as hydrogen bond donor and as acceptor in the docking study; which may be an outcome of the electron polarization due the coordination of amino groups to the metal centers.
What is the opinion of the authors: can a single interaction with the BCL-XL protein explain the cytotoxic activity of the compounds?
Reply: Docking score depends on the docking affinity of a compound with the target protein (10.3390/molecules23081899); which is not directly proportional to the number of interactions. Compound 1 interacted with the active sites of BCL-XL protein having Glu129(B) residue; while compound 2 interacted with the Phe105(B) residue. The docking score of compound 1 for BCL-2 and BCL-XL proteins were -99.92 and -86.29, respectively; whereas for compound 2, the scores are -59.78 and -62.59, respectively. From the docking score, it is evident that a single interaction with the BCL-XL protein is also effective in inducing apoptosis in DL cancer cells. The interaction of the compounds with BCL-XL protein indicates that the compounds are fitted in the active sites; thereby inhibiting its activity and resulting in apoptotic cell death (10.2174/1570163817999200729122753).
Lines 414-415: The phrase "The lack of any resonances at >8 ppm in the spectrum of 1 clearly demonstrates the absence of acidic protons of carboxylate groups" should be reformulated, because the carboxylate has no proton, the carboxyl group has a proton.
Reply: We have corrected the sentence in the revised manuscript.
If the authors highlight and explain the absence in the NMR spectrum of the signal of the proton from the carboxyl group, how do they explain the absence of the signals corresponding to the protons from the amine groups? By proton mobility?
Reply: The signals corresponding to the protons from the amine groups in aromatic ligands usually arises at 2.1-2.3 ppm (https://doi.org/10.1007/s12039-021-02022-0); which have merged with the signals of DMSO-d6 in the compounds 1 and 2. We have now revised the NMR section of the manuscript accordingly.
Reviewer 3 Report
In this paper, the authors reported the synthesis and the crystal structure of two Zn(II) coordination polymers, [Zn2Cl2(H2O)2(μ-4-aminobenzoate)2]n (compound 1) and [ZnCl2(μ-3-aminopyridine)2]n (compound 2).
The compounds were characterized suitably using various measurements such as FT-IR spectra, UV-Vis spectra, elemental analysis, 1H-NMR and TGA.
They reported the existence of interesting non-covalent interaction Type I Cl..Cl
and dual electron donor-acceptor topologies of aromatic rings in the crystal structure of the compounds.
The results of crystal structure analysis were supported by theoretical calculations.
They also reported the cytotoxicity and apoptosis assays of the compound 1 and compound 2, and the compounds induced significant concentration dependent cytotoxicity in Dalton’s lymphoma cancer cell, but negligible cytotoxicity in normal healthy peripheral blood mononuclear cell.
I think that the results reported in this paper are useful to develop not only the field of non-covalent interaction but also the field of drug for cancer therapy.
There are some minor comments as follows.
1. Figure S1, figure 2b
The number should be added to the atomic symbol.
O3, C5…..
2. Line 305, line 348
4-AmBz ligand
2 are
3. Line 413, line 508
Figure S6 → Figure S7. ?
Figure 8b→ Figure 5b. ?
4. Line 581
that ?
Author Response
First, we would like to thank this reviewers for his/her careful reading of the manuscript, corrections and suggestions. Our "point-by-point" responses follow:
In this paper, the authors reported the synthesis and the crystal structure of two Zn(II) coordination polymers, [Zn2Cl2(H2O)2(μ-4-aminobenzoate)2]n (compound 1) and [ZnCl2(μ-3-aminopyridine)2]n (compound 2). The compounds were characterized suitably using various measurements such as FT-IR spectra, UV-Vis spectra, elemental analysis, 1H-NMR and TGA. They reported the existence of interesting non-covalent interaction Type I Cl..Cl and dual electron donor-acceptor topologies of aromatic rings in the crystal structure of the compounds. The results of crystal structure analysis were supported by theoretical calculations. They also reported the cytotoxicity and apoptosis assays of the compound 1 and compound 2, and the compounds induced significant concentration dependent cytotoxicity in Dalton’s lymphoma cancer cell, but negligible cytotoxicity in normal healthy peripheral blood mononuclear cell.
I think that the results reported in this paper are useful to develop not only the field of non-covalent interaction but also the field of drug for cancer therapy.
There are some minor comments as follows.
- Figure S1, figure 2b. The number should be added to the atomic symbol.
O3, C5…..
Reply: We have incorporated the atom numbers in Figure S1 and Figure 2b
- Line 305, line 348 4-AmBz ligand 2 are
Reply: We have corrected the mistakes in the revised manuscript.
- Line 413, line 508
Figure S6 → Figure S7. ?
Figure 8b→ Figure 5b. ?
Reply: We have corrected the mistakes in the revised manuscript.
- Line 581 that ?
Reply: We have removed the additional ‘that’ from the sentence.
Reviewer 4 Report
This paper describes synthesis, X-ray structure, calculated bonding structure, and biological activity of two Zn complexes with aminobenzoic acid and aminopyridine ligands. Each subject is described in detail, and may be correct. However, there are serious defects. Several reports previously have appeared on the synthesis, structure, and biological activity of related Zn complexes with the same ligands, examples of which are noted below. Unfortunately, these works are not referred, discussed, nor compared in this paper. For docking study, the reason why this protein was selected is not described. No experimental evidence is provided that this can be a target. This paper is premature for a scientific publication, and this referee cannot recommend publication in Crystals in its present form.
Journal of the Korean Chemical Society 2013, Vol. 57, No. 6 712.
Journal of Food Protection, Vol. 72, No. 4, 2009, Pages 791–800.
Asian Journal of Chemistry Vol. 20, No. 8 (2008), 5827-5833.
Author Response
First of all we would like to thank this reviewer for his/her careful reading of the manuscript, corrections and suggestions. Our "point-by-point" responses follow:
This paper describes synthesis, X-ray structure, calculated bonding structure, and biological activity of two Zn complexes with aminobenzoic acid and aminopyridine ligands. Each subject is described in detail, and may be correct. However, there are serious defects. Several reports previously have appeared on the synthesis, structure, and biological activity of related Zn complexes with the same ligands, examples of which are noted below. Unfortunately, these works are not referred, discussed, nor compared in this paper.
Reply: We have now cited a few references in the revised manuscript. The papers referred by the reviewer are related to the antimicrobial activities of Zn compounds. Therefore we cannot compare their results of antimicrobial studies with our anticancer activities.
For docking study, the reason why this protein was selected is not described. No experimental evidence is provided that this can be a target.
Reply: We have mentioned the reason for selecting BCL family proteins for docking in the revised manuscript.
This paper is premature for a scientific publication, and this referee cannot recommend publication in Crystals in its present form.
Reply: Thank you for your opinion. In this case, we agree with the other four referees that this manuscript deserves publication
Reviewer 5 Report
Crystals-2216953-peer-review-v1
Comments to the Authors
Manuscript number Crystals-2216953-peer-review-v1 with the title “Unconventional Dual Donor-Acceptor Topologies of Aromatic Rings in Amine-based Polymeric Tetrahedral Zn(II) Compounds involving Unusual Non-Covalent Contacts: Antiproliferative Evaluation and Theoretical Studies” presents the synthesis characterization of two coordination polymers of Zn(II) viz. [Zn2Cl2(H2O)2(μ -4-AmBz)2]n (1) and [ZnCl2(μ -3-AmPy)2]n (2) (4-AmBz = 4-aminobenzoate, 3-AmPy = 3-aminopyridine). The crystal structure analyses of the polymers revealed the presence of non-covalent anion–π, π-stacking and unusual NH2(amino)⋯π interactions. The structures should be verified again in the CCDC database because I know for sure that one of them it is known, since 2018.
Molecular electrostatic potential (MEP) surface analysis show that the MEP values over the centre of the aromatic rings of coordinated 4-AmBz and 3-AmPy moieties are positive on one side and negative on the other side that fact confirming the dual non-covalent donor-acceptor topologies of the aromatic rings and explaining the concurrent formation of unusual non-covalent NH2···π and anion–π interactions.
The others studies: DFT calculations, QTAIM and NCI plot index brings out that among various non-covalent contacts involved in the crystal packing of the compounds, H-bonds in compound 1 and π-interactions (NH2···π, π-π, anion–π) in compound 2 are significant. In vitro antiproliferative evaluation of the compounds in Dalton’s lymphoma (DL) cancer cells revealed that the compounds induce concentration dependent cytotoxicity in DL cells with minimum effects in normal healthy PBMC cells.
The manuscript should have some MAJOR revisions before publishing.
Here are some questions and advices for the authors:
-The crystal structure for compound 2 is already known (that meaning that only one structure is new). Maybe should be cited.
-There are many typing errors and also the quality of the figures can be improved.
-The Molecular docking simulation part and the cytotoxicity and apoptosis analysis are interesting.
If the manuscript would have some Major revision before publishing, it will be interesting for the readers of the Crystals.
Author Response
First of all, we would like to thank this reviewer for his/her careful reading of the manuscript, corrections and suggestions. We have revised the manuscript accordingly. Our responses follow:
Manuscript number Crystals-2216953-peer-review-v1 with the title “Unconventional Dual Donor-Acceptor Topologies of Aromatic Rings in Amine-based Polymeric Tetrahedral Zn(II) Compounds involving Unusual Non-Covalent Contacts: Antiproliferative Evaluation and Theoretical Studies” presents the synthesis characterization of two coordination polymers of Zn(II) viz. [Zn2Cl2(H2O)2(μ -4-AmBz)2]n (1) and [ZnCl2(μ -3-AmPy)2]n (2) (4-AmBz = 4-aminobenzoate, 3-AmPy = 3-aminopyridine). The crystal structure analyses of the polymers revealed the presence of non-covalent anion–π, π-stacking and unusual NH2(amino)⋯π interactions. The structures should be verified again in the CCDC database because I know for sure that one of them it is known, since 2018.
Molecular electrostatic potential (MEP) surface analysis show that the MEP values over the centre of the aromatic rings of coordinated 4-AmBz and 3-AmPy moieties are positive on one side and negative on the other side that fact confirming the dual non-covalent donor-acceptor topologies of the aromatic rings and explaining the concurrent formation of unusual non-covalent NH2···π and anion–π interactions.
The others studies: DFT calculations, QTAIM and NCI plot index brings out that among various non-covalent contacts involved in the crystal packing of the compounds, H-bonds in compound 1 and π-interactions (NH2···π, π-π, anion–π) in compound 2 are significant. In vitro antiproliferative evaluation of the compounds in Dalton’s lymphoma (DL) cancer cells revealed that the compounds induce concentration dependent cytotoxicity in DL cells with minimum effects in normal healthy PBMC cells.
The manuscript should have some MAJOR revisions before publishing.
Here are some questions and advices for the authors:
-The crystal structure for compound 2 is already known (that meaning that only one structure is new). Maybe should be cited.
Reply: We have now cited the previously reported structure in the revised manuscript.
-There are many typing errors and also the quality of the figures can be improved.
Reply: We have done necessary modifications.
-The Molecular docking simulation part and the cytotoxicity and apoptosis analysis are interesting.
Reply: Thank you for your positive feedback.
If the manuscript would have some Major revision before publishing, it will be interesting for the readers of the Crystals.
Reply: We have done our best to improve the manuscript
Round 2
Reviewer 4 Report
Corrected version seems to be publishable.
Author Response
Thank you for the second reading of the manuscript recommending publication
Reviewer 5 Report
Dear Authors
Thank you for your answers. I have one more question for you!?
The structure of the compound 2 is like in the scheme below. I don’t understand why you insist in correcting the number of hydrogen. Can you correct in the supplementary files. Thank you for your patience and for your valuable work! Best regards!

Author Response
We thank this referee for his/her second reading of the manuscript and comment. Our reply follows:
Comment: The structure of the compound 2 is like in the scheme below. I don’t understand why you insist in correcting the number of hydrogen. Can you correct in the supplementary files. Thank you for your patience and for your valuable work!
Reply: This has been done and a new ESI has been uploaded